JCB Journal of Cell Biology

# ER export via SURF4 uses diverse mechanisms of both client and coat engagement

Julija Maldutyte[1] , Xiao-Han Li[1,2] , Natalia Gomez-Navarro[1] , Evan G. Robertson[2] , and Elizabeth A. Miller[1,2]

**Protein secretion is an essential process that drives cell growth and communication. Enrichment of soluble secretory proteins into ER-derived transport carriers occurs via transmembrane cargo receptors that connect lumenal cargo to the cytosolic COPII coat. Here, we find that the cargo receptor, SURF4, recruits different SEC24 cargo adaptor paralogs of the COPII coat to export different cargoes. The secreted protease, PCSK9, requires both SURF4 and a co-receptor, TMED10, for export via SEC24A. In contrast, secretion of Cab45 and NUCB1 requires SEC24C/D. We further show that ER export signals of Cab45 and NUCB1 bind co-translationally to SURF4 via a lumenal pocket, contrasting prevailing models of receptor engagement only upon protein folding/maturation. Bioinformatics analyses suggest that strong SURF4-binding motifs are features of proteases, receptor-binding ligands, and Ca²⁺-binding proteins. We propose that certain classes of proteins are fast-tracked for rapid export to protect the health of the ER lumen.**

## Introduction

Most secreted and membrane proteins enter the secretory pathway at the endoplasmic reticulum (ER) soon after translation begins. Upon docking of the ribosome at the ER, the nascent polypeptide chain engages with translocation machinery. Membrane proteins integrate into the ER membrane via partitioning of their transmembrane domains into the bilayer, whereas soluble proteins are released into the ER lumen. In both cases, the nascent protein acquires appropriate post-translational modifications (e.g., glycosylation) and quaternary structure, often aided by ER chaperones. According to current models, after secretory proteins are fully matured, folded, and have undergone ER quality control, they are exported via transport vesicles made by the COPII coat machinery (Barlowe and Helenius, 2016; Dancourt and Barlowe, 2010).

Enrichment of proteins into COPII vesicles is driven by the SEC24 subunit of the coat, which binds to ER export motifs on membrane proteins. Soluble secretory proteins require cargo receptors to connect lumenal proteins to SEC24. Cargo receptors must thus interact simultaneously with their clients and SEC24. Mammals express four SEC24 orthologs: SEC24A and SEC24B are 75% identical to each other, with SEC24C and SEC24D sharing 66% similarity (Mancias and Goldberg, 2008). Each SEC24 possesses multiple sites of engagement with cargo. All four homologs share a universally conserved cargo binding site, termed the

B-site, with SEC24A/B and SEC24C/D also containing sites specific to each homologous pair (Chatterjee et al., 2021; Mancias and Goldberg, 2008). The full spectrum of proteins recognized by each of these cargo-binding sites remains unknown.

The conserved ER export receptor, SURF4 (Erv29 in yeast) (Belden and Barlowe, 2001; Otte and Barlowe, 2004), drives traffic of a diverse set of secretory cargo proteins (Saegusa et al., 2018, 2022; Yin et al., 2018; Lin et al., 2020; Wang et al., 2020; Devireddy and Ferguson, 2021; Huang et al., 2021; Gomez-Navarro et al., 2022; Tang et al., 2022a, 2022b, 2023). Recognition of SURF4 clients appears to occur via two mechanisms: Cardin-Weintraub (CW) motifs on clients bind via a putative lumenal α-helix (Tang et al., 2022a, 2022b) and N-terminal ER-ESCAPE (ER-Exit by Soluble Cargo using Amino-terminal Peptide Encoding) signals are recognized by an unknown mechanism (Devireddy and Ferguson, 2021; Gomez-Navarro et al., 2022; Yan et al., 2022; Yin et al., 2018). SURF4 itself appears to interact with the B-site of SEC24A (Gomez-Navarro et al., 2022), consistent with an established role for SEC24A in secretion of the SURF4 client, proprotein convertase substilisin/kexin type 9 (PCSK9) (Chen et al., 2013; Emmer et al., 2018; Gomez-Navarro et al., 2022).

Here, we sought to dissect the mechanisms of SURF4-mediated secretion. Surprisingly, unlike PCSK9, SURF4

[1]MRC Laboratory of Molecular Biology, Cambridge, UK;   [2]Division of Molecular, Cell and Developmental Biology, School of Life Sciences, University of Dundee, Dundee, UK.

Correspondence to Elizabeth A. Miller: emiller@mrc-lmb.cam.ac.uk, emiller003@dundee.ac.uk

N. Gomez-Navarro's current affiliation is Section on Structural and Chemical Biology of Membrane Proteins, Neurosciences and Cellular and Structural Biology Division, Eunice Kennedy Shriver National Institute of Child Health and Human Development, National Institutes of Health, Bethesda, MD, USA.

cargoes Cab45 and NUCB1 use SEC24C/D for ER export, raising the question of how different lumenal clients influence cargo adaptor specificity. In the case of PCSK9, we show a requirement for a co-receptor, TMED10, that likely contributes to this specificity. We map the site of ER-ESCAPE motif binding to a highly conserved lumenal domain of SURF4. Contrary to prevailing views of folding-dependent cargo-binding, we found that Cab45 and NUCB1 interact with SURF4 co-translationally, dependent on signal peptide (SP) cleavage which exposes the ER-ESCAPE motif. We propose that this co-translational mode of interaction ensures the rapid export of $Ca^{2+}$-binding proteins to prevent $Ca^{2+}$ sequestration and disruption to $Ca^{2+}$ homeostasis in the ER.

## Results

### SURF4 differentially recruits COPII cargo adaptor subunits for export of distinct soluble cargoes

We previously showed that Cab45 and NUCB1 depend on SURF4 for efficient secretion, similar to PCSK9 (Gomez-Navarro et al., 2022; Emmer et al., 2018). In the case of PCSK9, traffic was specifically dependent on SEC24A (Chen et al., 2013), likely via its B-site (Gomez-Navarro et al., 2022). We, therefore, tested whether Cab45 and NUCB1 also exclusively use SEC24A for secretion. Surprisingly, neither cargo was affected in SEC24A KO HEK-293TREx cells (Fig. 1, A and B). Secretion of Cab45 was also unaffected when SEC24B (homologous to SEC24A) was knocked out, either alone (Fig. S1 A) or in combination with SEC24A transient knockdown (Fig. 1, C and D). In contrast, Cab45 secretion was reduced when both SEC24C and SEC24D (homologous to each other) were downregulated (Fig. 1, C and D) but not in SEC24C or SEC24D KO alone (Fig. S1 A). Since we had previously shown that the SEC24 B-site inhibitor, 4-phenylbutyrate (4-PBA), abrogated secretion of Cab45 and NUCB1 (Gomez-Navarro et al., 2022), we tested whether the SEC24C B-site drives the SEC24C–SURF4 interaction. Using an in-cell protein–protein interaction assay based on reconstitution of luciferase activity (NanoBiT; NanoLuciferase Binary Technology) (Fig. 1 E), we saw robust interaction upon co-expression of SEC24C and SURF4 (Fig. 1 F; WT). In addition to the conserved B-site, SEC24C also has a separate cargo-binding site that binds IxM motifs (here termed IxM-site) (Mancias and Goldberg, 2008; Fig. 1 G). Mutations were introduced into these sites and assessed for stability (Fig. S1 B), then tested in the NanoBiT assay. Only the B-site mutant showed reduced interaction with SURF4 (Fig. 1 F). Moreover, increasing concentrations of the B-site inhibitor 4-PBA significantly reduced SEC24C–SURF4 interaction (Fig. 1 H). Our NanoBit results are consistent with a SURF4-–SEC24C interaction driven by the conserved B-site, similar to that observed previously for SEC24A (Gomez-Navarro et al., 2022).

We next aimed to identify the signals on SURF4 that are recognized by SEC24. It has previously been proposed that SURF4 engages SEC24 via a FF motif, located close to the C-terminus (Devireddy and Ferguson, 2021). Although such a motif might be recognized by the B-site based on structural and biochemical studies (Ma et al., 2017; Nie et al., 2018; Nufer et al., 2002), recent structural predictions of SURF4 suggested that

these residues lie within the final transmembrane domain, and thus are unlikely to be exposed for SEC24 interaction (Fig. S1 C). We therefore undertook extensive site-directed mutagenesis of SURF4 guided by structural predictions and conservation across species (Fig. S1 C shows locations of mutations on a SURF4 topology model). Our mutagenesis approach identified a number of candidate mutants that showed reduced secretion of Cab45 with no impact on SURF4 stability (Fig. S1 D and Table S1). Three mutants that were ER-retained (Fig. S2) were tested in the NanoBiT assay for SEC24A and SEC24C interaction. These mutations were in predicted cytoplasmic regions: a loop between transmembrane helices 4 and 5, which we termed the Phe-loop; and two mutations in the C-terminal tail (Fig. 2 A). The Phe-loop contains a conserved phenylalanine residue, similar to the signal identified on yeast Erv14 (Pagant et al., 2015); the C-terminal region contains two potential motifs: a diacidic (DE) motif and a canonical ΦC motif (EW) at the very C-terminus, both of which would be predicted to bind the B-site (Mossessova et al., 2003; Ma et al., 2017). Alanine substitutions were introduced at these sites into the LgBiT-SURF4 construct, then tested for stability (Fig. S1 E) and interaction with SmBiT-SEC24 (Fig. 2 B). The Phe-loop mutation reduced interaction with SEC24A but not with SEC24C (Fig. 2 B). In contrast, both C-terminal DE and EW mutants significantly perturbed SURF4 interaction with SEC24C with minimal impact on SEC24A binding (Fig. 2 B). Diacidic and ΦC signals bind the B-site across species (Mancias and Goldberg, 2008; Miller et al., 2003; Mossessova et al., 2003; Ma et al., 2017). The Phe-based signal has only been defined in yeast and is recognized by the D-site, identified by mutagenesis of yeast Sec24, the yeast SEC24A ortholog (Pagant et al., 2015; Powers and Barlowe, 2002). We identified a surface pocket equivalent to the D-site on SEC24A (Fig. 2 C) and introduced alanine substitutions in this region to our NanoBiT constructs (Fig. S1 B). Indeed, the D-site mutant displayed reduced interaction with SURF4, albeit to a lesser extent than B-site mutants (Fig. 2 D). Together, the NanoBit interaction data led us to hypothesize that SURF4 differentially binds SEC24 paralogs via distinct modes: it interacts with the B-site of SEC24C directly via diacidic and/or ΦC signals in its C-terminal domain, and binds the D-site of SEC24A via its Phe-loop. Since the adjacent B-site on SEC24A is also important for SURF4 interaction (Gomez-Navarro et al., 2022; Fig. 2 D), we propose that this site might bind a co-receptor. Similar coincidence detection has been shown to drive ER export in yeast, where Sec24 simultaneously binds a polytopic membrane protein, Yor1, and its receptor, Erv14, via the B- and D-sites respectively (Pagant et al., 2015) (Fig. 2 E).

### TMED10 is necessary for SEC24A-mediated PCSK9 exit from the ER

To identify potential co-receptors that work in combination with SURF4, we searched for proteins with predicted shared phenotypes, as determined from CRISPR screens and bioinformatics analyses, otherwise known as co-essentiality mapping (Wainberg et al., 2021). In DepMap and Co-essentiality Browser databases (Gillani et al., 2021; Wainberg et al., 2021), SURF4 clusters with COPII machinery and a number of ER membrane proteins (Fig. S3 A). Two known cargo receptors,

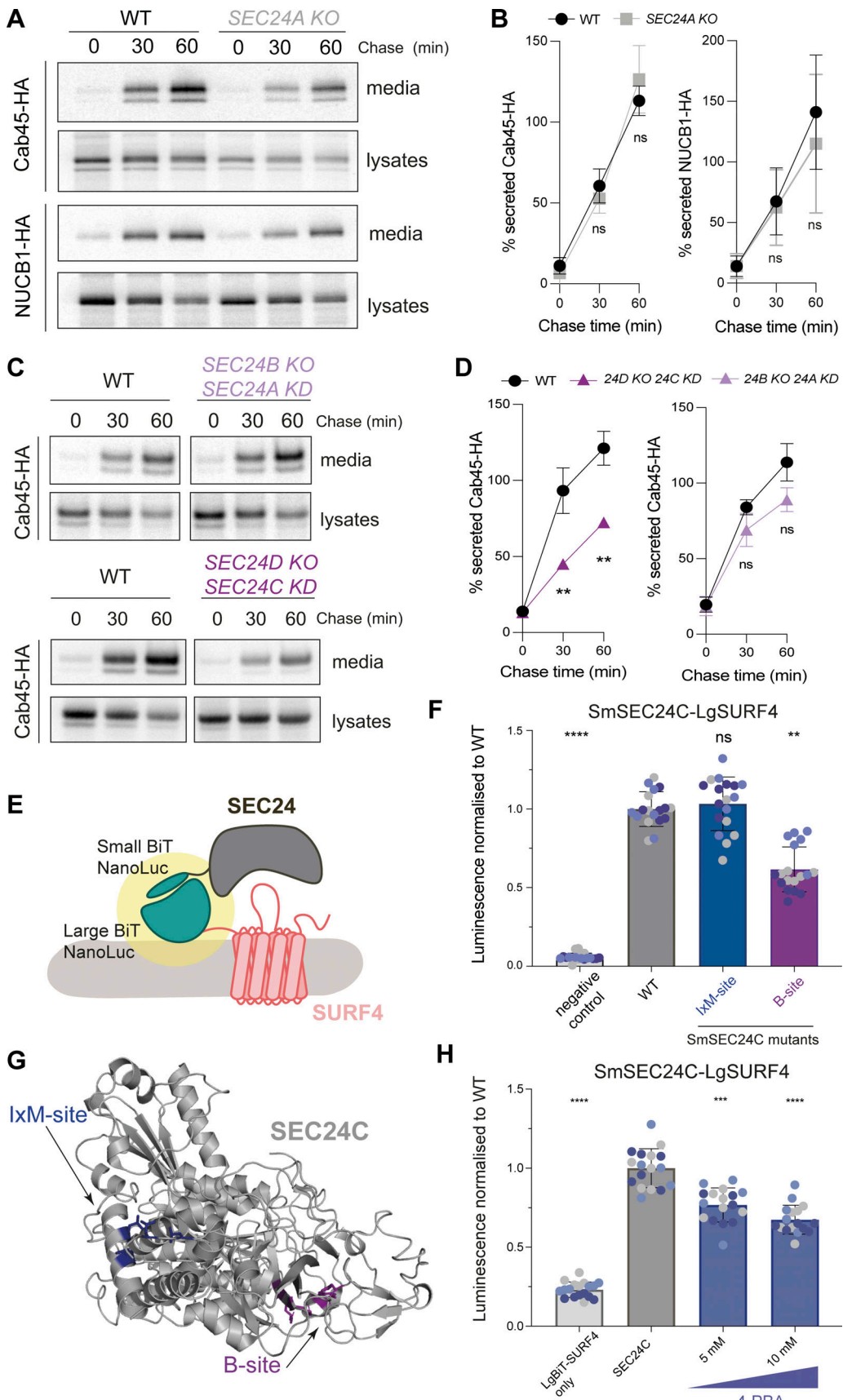

Figure 1. **SEC24C/D B-site drives Cab45 secretion. (A)** Radiolabeled pulse-chase experiment testing Cab45 and NUCB1 secretion in WT and SEC24A KO cells. Transiently transfected HA-tagged proteins were immunoprecipitated from media and lysates at indicated time points after [35S]-Met/Cys addition and

detected by SDS-PAGE and phosphorimage analysis. **(B)** Protein secretion shown in A was quantified from three independent experiments. **(C)** Radiolabeled pulse-chase experiment testing Cab45 secretion in WT and KO/KD cells as indicated. **(D)** Quantification of secretion experiments shown in B; $n = 3$. **(E)** Diagram of NanoBiT complementation assay to assess SURF4-SEC24 interactions. **(F)** NanoBiT luminescence measured upon co-expression of LgBiT-SURF4 with the indicated SmBiT-SEC24C mutants, normalized to WT values. Negative control was a smallBiT-PRKACA (activation of protein kinase A) fusion that controls for Small-BiT background complementation. **(G)** Crystal structure of SEC24C (PDB ID: 3EH2) showing the two binding sites tested in the NanoBiT assay. **(H)** NanoBiT luminescence was measured upon co-expression of LgBiT-SURF4 and SmBiT-SEC24C in the presence of the B-site-occluding small molecule, 4-PBA, normalized to WT. Negative control was largeBiT-PKAR2A (protein kinase cAMP-dependent type II regulatory subunit alpha) that controls for non-specific large-BiT binding. Six technical replicates were used in each of the three independent NanoBiT biological replicates, as indicated by differential coloring within superplots. Triangles represent mean and error bars represent SD. Statistical tests were one-way ANOVA with Dunnett's correction for multiple testing. Data distribution was assumed to be normal but this was not formally tested. ns = not significant. * = P value <0.033, ** = P value <0.002, *** = P value <0.0002, **** = P value <0.0001. Source data are available for this figure: SourceData F1.

TMED2 and TMED10 (Bare et al., 2023; Fujita et al., 2011; Satpute-Krishnan et al., 2014; Strating and Martens, 2009; Zavodszky and Hegde, 2019), cluster closest to SURF4, suggesting conserved phenotypes when each of these proteins are mutated. Moreover, OpenCell analysis, which used genome-scale GFP-tagging and immunopurification-mass spectrometry studies (Cho et al., 2022), suggests that TMED2 and TMED10 physically interact with SURF4. To test whether TMED2 and TMED10 might function as SURF4 co-receptors in SEC24A-mediated export from the ER, we knocked them down (Fig. S3 B) and measured the effect on the SEC24A-specific cargo, PCSK9. PCSK9 secretion was markedly reduced in TMED10 but not TMED2 knockdown cells (Fig. 3, A and B), whereas SEC24C/D/SURF4 cargoes, Cab45 and NUCB1, were unaffected (Fig. S3, C and D). Moreover, the knockdown of TMED10 significantly reduced SURF4–SEC24A interaction (Fig. 3 C), suggesting the ternary complex is important for SURF4 engagement with SEC24A. In contrast, neither TMED2 nor LMAN1 (another abundant cargo receptor that exports glycoproteins [Appenzeller-Herzog et al., 2005, 2004; Nufer et al., 2003; Zheng et al., 2013]) impacted SURF4–SEC24 interaction (Fig. 3 C), and the SURF4–SEC24C interaction was unaffected by TMED10, TMED2, or LMAN1 knockdown (Fig. S3 E).

To test whether TMED10 directly engages with PCSK9, we performed dithiobis-succinimidyl propionate (DSP) crosslinking and co-immunoprecipitation (co-IP) experiments from TMED10 KO cells transiently co-transfected with PCSK9-V5 and HA-TMED10. IP of PCSK9 recovered TMED10, detected with an antibody against the cytosolic tail (Fig. 3 D, upper panel). Reciprocally, anti-HA IP of TMED10 co-precipitated PCSK9, with the higher molecular weight precursor the dominant form recovered (Fig. 3 D, lower panel). We next tested TMED10 mutants deleted for their lumenal coiled-coil or GOLD domains. The GOLD domain has been previously implicated in cargo interactions (Anantharaman and Aravind, 2002; Zavodszky and Hegde, 2019), whereas the coiled-coil domain drives oligomerization of TMED family proteins (Ciufo and Boyd, 2000; Emery et al., 2000). In both IP orientations, the coiled-coil deletion mutant co-immunoprecipitated with PCSK9-V5, again predominantly with the immature precursor form. In contrast, upon GOLD domain deletion, TMED10 recovery with V5-PCKS9 was dramatically reduced. In the reciprocal IP, the precursor form of PCSK9 was poorly recovered, and instead, the mature form was more abundant (Fig. 3 D). One possibility is that the mature PCSK9 recovered is bound to SURF4, which is presumably

within the holo-complex that drives ER export and therefore also recovered in the TMED10 IP. None of the TMED10 constructs interacted with endogenous Cab45 (Fig. S3 F). We were unable to directly test TMED10 engagement at the B-site of SEC24A; LgBiT-TMED10 did not function in the NanoBiT assay, presumably because the luciferase construct sterically occludes its ER export signal. Moreover, we could not detect an interaction by co-IP, likely due to the transient nature of TMED10-SEC24A interaction. Based on previous detailed biochemical and structural studies (Ma et al., 2017; Nie et al., 2018; Nufer et al., 2002; Strating and Martens, 2009), we propose that TMED10 is likely to engage the SEC24A B-site via either its FF or C-terminal IE motif. Overall, our data thus far suggest that the GOLD domain of TMED10 binds to the precursor form of PCSK9. Subsequent self-cleavage of PCSK9 would reveal the ER-ESCAPE motif for SURF4 engagement, with co-incident recognition of the PCSK9/SURF4/TMED10 assembly by the B- and D-sites of SEC24A (Fig. 3 E). In contrast, Cab45 recognition is more canonical in that a simple interaction network bridges Cab45 in the lumen through SURF4 to the B-site of SEC24C (Fig. 3 E).

## ER-ESCAPE motif is bound at a putative ER lumen-facing pocket of SURF4

To dissect the mechanism by which SURF4 recruits proteins for ER export, we sought to detect direct interaction between client and export receptor using in vitro translation (IVT) and site-specific crosslinking. Cab45 contains an ER-ESCAPE motif (Yin et al., 2018; Devireddy and Ferguson, 2021; Gomez-Navarro et al., 2022; Yan et al., 2022) that is revealed at the N-terminus after SP cleavage and is immediately followed by an N-glycosylation site (Fig. 4 A). Using amber STOP codon suppression, we introduced a photo-crosslinkable amino acid p-benzoyl-Lphenylalanine (Bpa) (Chin et al., 2002) into in vitro translated Cab45-HA constructs at positions +3 (A39) and +4 (N40) downstream of the SP cleavage site (Fig. 4 A). Translocation of these substrates into the ER was induced by the inclusion of semipermeabilized cells in the translation reaction. For this purpose, we used HEK-293TREx SURF4 KO cells that stably express FLAG-tagged SURF4 (Fig. S4 A). Translation/translocation reactions were UV-crosslinked to create covalent adducts between Bpa and adjacent amino acids on nearby proteins, and then Cab45-HA and FLAG-SURF4 were immunoisolated and analyzed by SDS-PAGE. Crosslinked products of equal molecular weight were detected with both HA and FLAG IPs when Bpa was placed at position N40 (N40*). The size of this

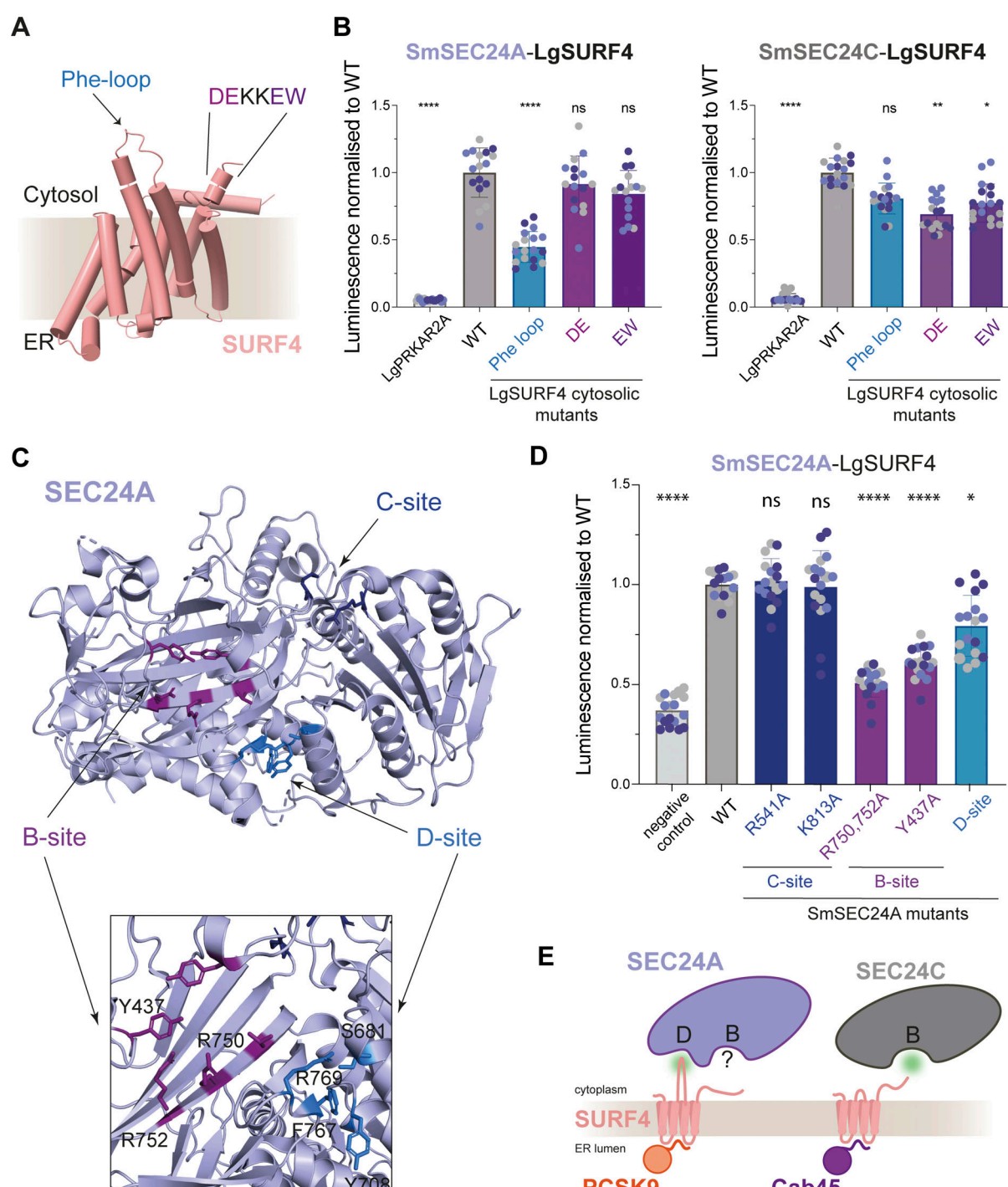

Figure 2. **Mutagenesis reveals complexity of SURF4 interactions with SEC24 paralogs. (A)** AlphaFold2 model of SURF4, highlighting cytosolic regions important for SURF4-dependent cargo secretion identified by mutagenesis. **(B)** Luminescence values were measured upon co-expression of SmBiT-SEC24C or SmBiT-SEC24A with the indicated LgBiT-SURF4 mutants, normalized to WT values. **(C)** Crystal structure of SEC24A (PDB ID: 5VNO) showing the three binding sites tested in the NanoBiT assay; neighboring B- and D-sites are zoomed in. **(D)** NanoBiT complementation measured upon co-expression of LgBiT-SURF4 and the indicated SmBiT-SEC24A mutants, normalized to WT. **(E)** Model summarizing secretion dissected by NanoBiT and pulse-chase. SEC24A engages SURF4 via a D-site-cytosolic loop interaction, whereas the B-site is important for PCSK9 secretion but not for SURF4 engagement. SEC24C engages C-terminal motifs of SURF4 via its B-site to drive Cab45 export. Statistical tests were one-way ANOVA with Dunnett's correction for multiple testing. Data distribution was assumed to be normal but this was not formally tested. ns = not significant, * = P value <0.033, ** = P value <0.002, *** = P value <0.0002, **** = P value <0.0001. For each NanoBiT experiment, six technical replicates were used in each of the three independent biological replicates, as indicated by differential coloring within superplots. Triangles represent mean and error bars represent SD.

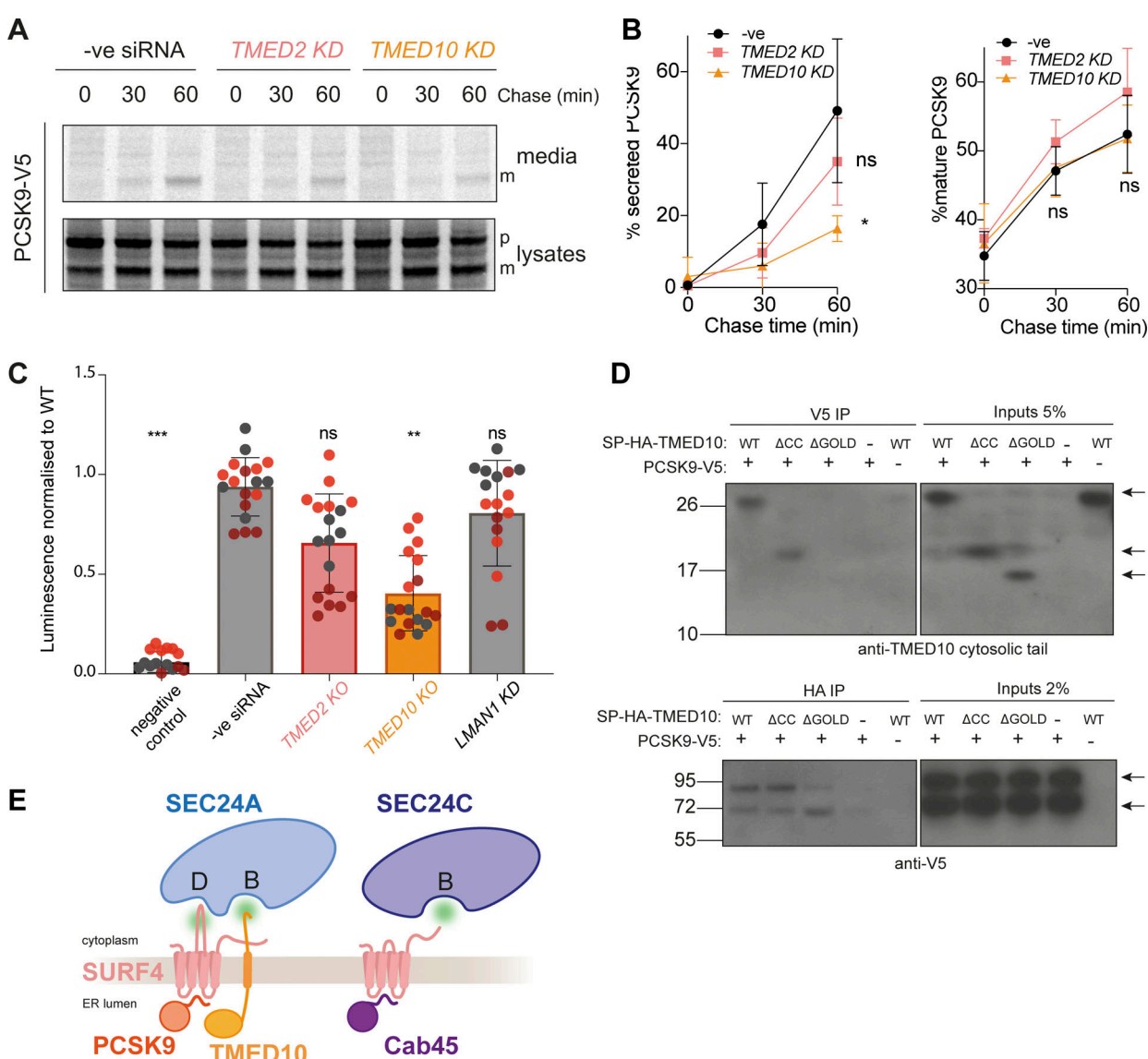

Figure 3. **TMED10 is required for efficient PCSK9 export. (A)** Radiolabeled pulse-chase of PCSK9 secretion. HEK-293TREx cells were transfected with the indicated siRNAs, and after 24 h were transfected with a plasmid expressing PCSK9-V5, then subjected to pulse-chase analysis the following day. **(B)** Protein secretion was quantified from three biological replicates. **(C)** SURF4/SEC24 double KO cells were transfected with the indicated siRNAs, then after 24 h co-transfected with SmBiT-SEC24A and LgBiT-SURF4 NanoBiT constructs. Luciferase luminescence values were normalised to WT. Triangles represent mean and error bars represent SD. For each NanoBiT experiment, six technical replicates were used in each of the three independent biological replicates, as indicated by differential coloring within superplots. Triangles represent the mean and error bars represent SD. **(D)** DSP-crosslinking co-immunoprecipitation of PCSK9-V5 and SP-HA-TMED10 WT, coiled-coil and GOLD domain deletion mutants. The indicated constructs were co-transfected in TMED10 KO cells (- means empty pcDNA3.1 vector was used). Cells were collected, treated with DSP and cleared lysates co-immunoprecipitated overnight. Antibody to detect TMED10 signal was against TMED10 cytosolic tail. **(E)** A cartoon model illustrating the two mechanisms by which SURF4 relays cargo recruitment to the inner COPII coat. Statistical tests were one-way ANOVA with Dunnett's correction for multiple testing. Data distribution was assumed to be normal but this was not formally tested. ns = not significant, * = P value <0.033, ** = P value <0.002, *** = P value <0.0002. Source data are available for this figure: SourceData F3.

product (∼75 kDa) was consistent with an adduct containing Cab45 (∼47 kDa) and SURF4 (∼28 kDa) (Fig. 4 B). Crosslinked products were not detected when Bpa was incorporated directly into the ER-ESCAPE motif (A39*), suggesting that the non-natural amino acid disrupted interaction with SURF4. Importantly, crosslinked Cab45-N40* product was abrogated when the ER-ESCAPE motif was mutated (Fig. 4 B). We note that Bpa incorporation at position N40 disrupts *N*-glycosylation but does not interfere with SP cleavage (Fig. S4, B and C). We have

previously shown that Cab45 secretion is unaffected when glycosylation is perturbed (Gomez-Navarro et al., 2022). Moreover, a Cab45/SURF4 product was also detected when Bpa was placed downstream of the glycosylation site (T44*; Fig. S4 D).

To identify the site on SURF4 that recognizes ER-ESCAPE motifs, we returned to our library of SURF4 mutants, focusing on predicted lumenal substitutions that impaired Cab45 secretion (Fig. S1 C and Table S1). Mutants with the most pronounced effect were M1 and M7–9. The M1 mutation partially maps onto a

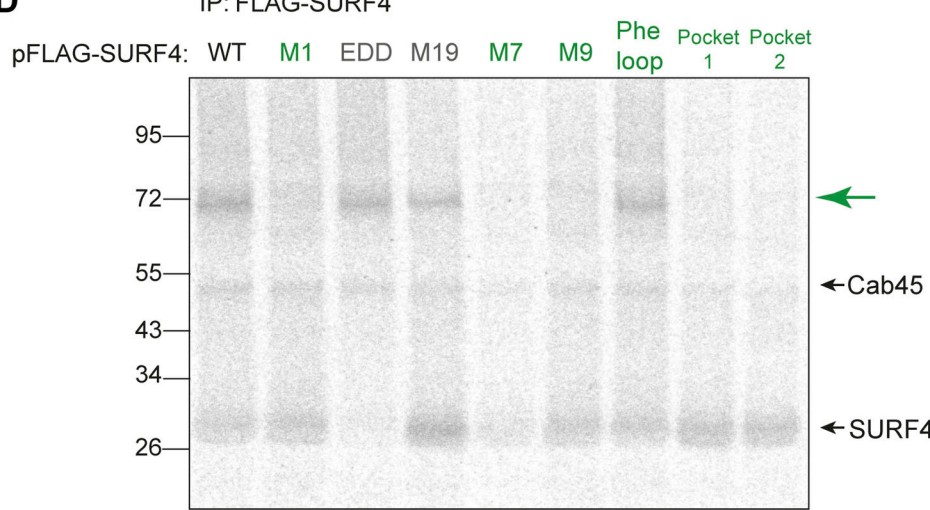

Figure 4.   **ER-ESCAPE binds a conserved lumenal pocket of SURF4. (A)** N-terminal amino acid sequence of Cab45. Numbering indicates amino acid positions from the N-terminus, including SP. An ER-ESCAPE motif (black box) follows a signal peptide cleavage site (arrow), followed by a putative *N*-glycosylation site. Green shading shows positions that were crosslinked to SURF4. **(B)** A site-specific photo-crosslinking experiment showing the dependence of Cab45-SURF4 interaction on ER-ESCAPE. HA and FLAG IPs were performed on UV-crosslinked semi-permeabilized cells. Green arrows indicate crosslinked species recovered by both Cab45-HA and FLAG-SURF4 IPs. Presumed migration of Cab45 species and SURF4 are indicated. **(C)** ER lumenal views of SURF4 AF2 structure predictions showing hydrophobicity, surface charge, and conservation scores (left to right). The long arrow points to the predicted pocket for ER-ESCAPE binding, the small arrow indicates another, smaller putative pocket. A lumenal α-helix previously shown to bind CW motifs, and M7-9 regions are circled. **(D)** IVT and site-specific photo-crosslinking of Cab45-N40* to SURF4 and its mutants. HEK-293TREx SURF4 KO cells were transfected with the indicated FLAG-SURF4 constructs 24 h prior to IVT. Further sample processing was performed as in B. SURF4 mutants in green are those that did not produce a cross-link with SURF4 (indicated by green arrow), and mutants in grey produced a crosslink. Source data are available for this figure: SourceData F4.

predicted negatively charged lumenal α-helix (Fig. 4 C and Fig. S1 C). This helix also contains a previously described recognition site for an alternative export signal, the CW motif (Tang et al., 2022a). AlphaFold2 structural predictions suggested that the M7–M9 mutations map to a highly conserved lumenal pocket that is lined with both negatively charged and hydrophobic residues (Fig. 4 C and Fig. S1 C). The geometry and chemical properties of this pocket would support the binding of the positively charged and hydrophobic ER-ESCAPE motif. We, therefore, tested the ability of these mutants to interact with Cab45 in the site-specific crosslinking assay by transiently transfecting them into SURF4 KO cells and translating Cab45-N40*. Neither M7 nor M9 SURF4 mutants crosslinked to Cab45-N40*, suggesting this region was essential for SURF4 binding to the ER-ESCAPE motif of Cab45 (Fig. 4 D). Moreover, when this predicted pocket was occluded with electrostatic bridges (Pocket 1 and 2 mutants, Fig. 4 D and Table S1), crosslinked products were lost, further supporting this pocket as the ER-ESCAPE recognition domain. In contrast, crosslinks were unaffected in a more distant lumenal mutant, M19, the cytosolic Phe-loop mutant, and a CW binding site mutant, EDD (Fig. 4 D).

## Cab45 and NUCB1 ER-ESCAPE is bound by SURF4 co-translationally

For many SURF4 cargoes, ER-ESCAPE motifs are revealed following SP cleavage. We sought to determine if this N-terminal positioning of the ER-ESCAPE motif is necessary for cargo to engage SURF4. We translated Cab45-N40* in the presence of a signal peptidase complex inhibitor, cavinafungin (CVF) (Estoppey et al., 2017). CVF treatment indeed inhibited SP cleavage (Fig. 5 A) and abrogated crosslinked SURF4–Cab45 products (Fig. 5 B). SP cleavage inhibition and loss of interaction with SURF4 was also achieved by introducing mutations near the SP cleavage site, predicted from previous studies to disrupt SP cleavage (Fig. S4, B, C, and E) (Paetzel, 2014). Finally, we confirmed that mutations that disrupt SP cleavage abrogate secretion of Cab45 (Fig. S4 F).

SP cleavage often occurs co-translationally, exposing the ER-ESCAPE motif early during synthesis. We therefore wondered if this permits co-translational interaction with SURF4. We tested this by programming the IVT/translocation reaction with truncated Cab45 mRNA that lacks a stop codon, thereby creating a ribosome-trapped nascent chain in a co-translationally stalled state. If Cab45 interacts co-translationally with SURF4, truncated constructs should preserve SURF4–Cab45 crosslinked adducts. We created a series of Cab45 truncations to monitor both the shortest nascent protein required for SURF4 engagement and to test prolonged interaction as nascent chain length increased. We tested constructs that incorporated Bpa at two sites: N40* and T44*, plus a glycosylation mutant (N40G) in the context of the T44* condition. Following IVT, crosslinking, and FLAG-SURF4 IP, we observed a series of Cab45 interaction products, suggesting co-translational engagement with SURF4. The earliest crosslinked product was visible when 125 amino acids of Cab45 were translated (Fig. 5 C). The Cab45 125-mer includes (C-term to N-term) ~30 amino acids in the ribosome exit tunnel, ~20 residues in the translocon, leaving ~75 amino

acids exposed to the lumen. Assuming the 36 amino acid-long SP is cleaved, this leaves an ~39-amino acid-long polypeptide available for interaction with SURF4, suggesting that steric accessibility of the nascent Cab45 polypeptide is the only limiting factor preventing early interaction with SURF4. The T44* construct behaved similarly, although the intensity of the cross-linked product was reduced relative to N40* (similar to full-length T44*), but abrogating glycosylation increased the interaction signal again, consistent with steric effects of glycosylation and/or competition with the glycosylation machinery. We also detected additional faint higher molecular weight crosslinks, one of which corresponds to a Cab45-TRAPα interaction (Fig. S4 G) and the other product unidentified (Fig. 5 C). We similarly detected co-translational binding between NUCB1 and SURF4 (Fig. S4 H).

## Properties of secretory proteins with strong ER-ESCAPE motifs

Our finding that both Cab45 and NUCB1 engage with SURF4 co-translationally is at odds with current thinking about ER export in the context of protein folding. Most proteins are thought to only engage with their export receptors once fully folded and released from ER chaperones (Dancourt and Barlowe, 2010; Barlowe and Helenius, 2016). We therefore sought to understand more about the properties that dictate SURF4 dependence. We compiled a dataset of secreted human proteins and developed an "ER-ESCAPE score" for each N-terminal sequence (following SP cleavage), with values assigned according to previous systematic experimental analysis of amino acid functionality within this motif (Yin et al., 2018). A fully optimal ER-ESCAPE motif would have a score of 15, whereas a fully non-optimal would score –15 (Fig. 6 A and Fig. S5 A). We note that this pipeline does not accurately capture values for proteins like PCSK9, which reveal their ER-ESCAPE following the removal of a propeptide.

With 1,988 proteins assigned an ER-ESCAPE score, we tested whether there was a statistical difference between median ER-ESCAPE scores of protein populations with different properties. Co-translational interaction with SURF4 implies that cargoes engage the receptor prior to folding. We reasoned that this would be detrimental for proteins with complex folding needs, which would require significant ER residence time. Such proteins would have poor ER-ESCAPE propensity. Indeed, annotated presence of disulfide bonds was significantly associated with lower ER-ESCAPE scores (Fig. 6 B). In contrast, it has previously been proposed that SURF4 engagement might be a "fast-track" out of the ER for proteins that undergo $Ca^{2+}$-dependent oligomerization (Yin et al., 2018). We therefore asked whether $Ca^{2+}$ binding was associated with higher ER-ESCAPE strength, finding a higher median ER-ESCAPE score for annotated $Ca^{2+}$-binding proteins (Fig. 6 C). Within the 228 $Ca^{2+}$-binding proteins, those with EF-Hand or multiple $Ca^{2+}$-binding sites show even higher ER-ESCAPE scores (Fig. S5, B and C). We next took an unbiased approach to search for properties associated with high and low ER-ESCAPE scores. Gene ontology (GO) term analysis revealed that the set of proteins with the strongest ER-ESCAPE scores (≥12) were strongly enriched for a number of categories of small secreted ligands, including cytokine/

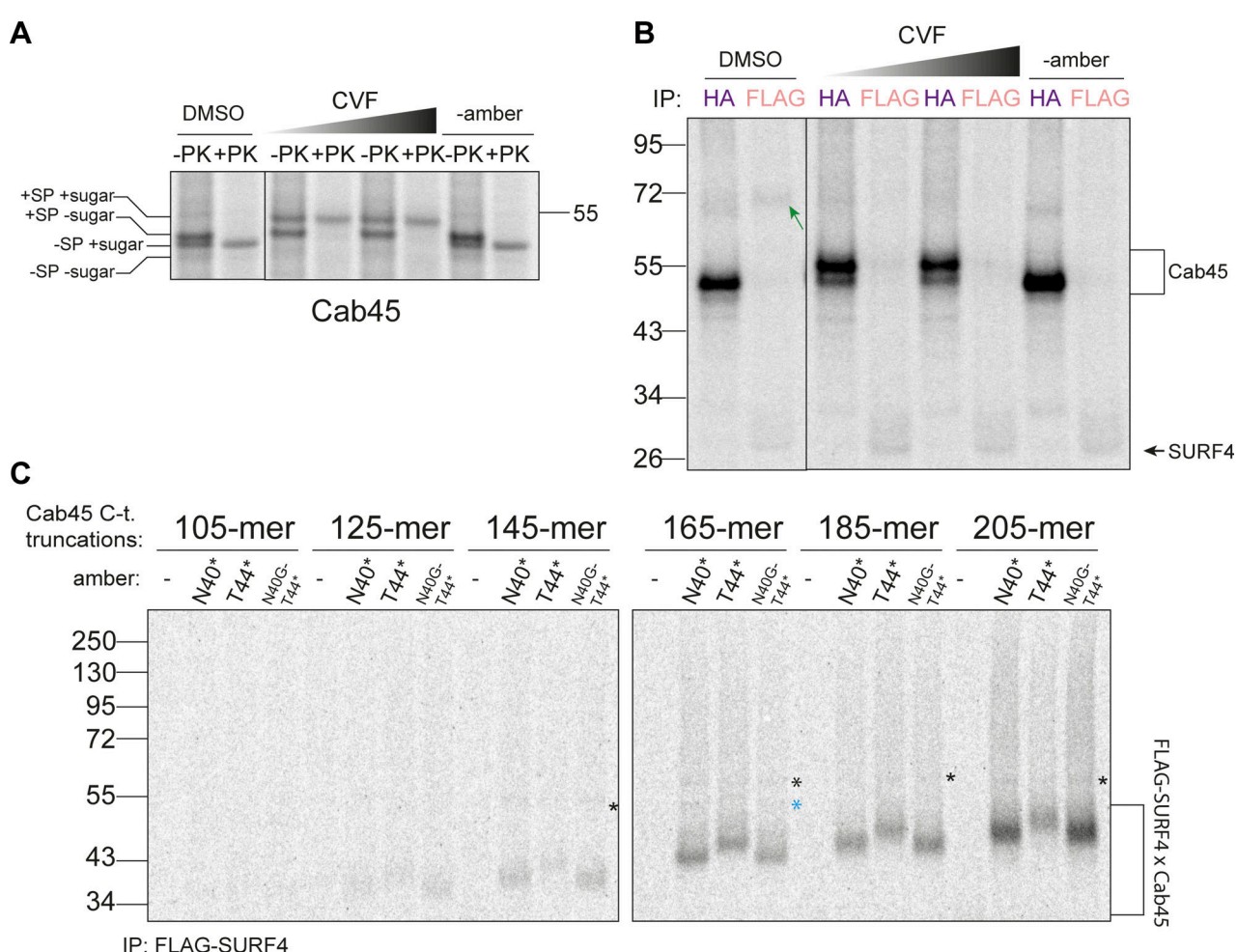

**Figure 5.   Cab45-SURF4 interaction is SP cleavage-dependent and occurs co-translationally. (A)** Proteinase K (PK) protection assay of Cab45-T44* or Cab45 with no amber suppression, treated with an increasing concentration of cavinafungin (CVF; 1 μM, 10 μM in DMSO [1% of IVT reaction volume]) or vehicle DMSO. Samples were taken from total IVT reactions. Positions of different species of Cab45 are indicated. **(B)** The same samples as in A but Cab45-HA and FLAG-SURF4 IPs were performed on UV-crosslinked semi-permeabilized cells. Green arrow indicates Cab45-SURF4 crosslink. **(C)** Cab45 C-terminal truncations were translated in vitro and photo-crosslinked to FLAG-SURF4. Following UV-crosslinking and FLAG-IP, cross-linked products were resolved by SDS-PAGE. * indicates a consistent cross-link that was later identified to be TRAPα (Fig. S4 G). Blue * indicates an unidentified crosslinked species. Source data are available for this figure: SourceData F5.

chemokines and hormones, as well as proteases (Fig. S5 D). We therefore compared ER-ESCAPE strength for the GO terms "receptor ligand activity" and "peptidase," finding significantly higher ER-ESCAPE scores for these populations (Fig. 6, D and E). In the population of proteins with strongly negative ER-ESCAPE scores, we found enrichment for large secretory cargo terms such as lipoproteins, extracellular matrix, as well as platelet-derived growth factor binding (Fig. S5 E). In agreement with this, apolipoproteinB (ApoB), a major constituent of lipoproteins, has an unfavorable ER-ESCAPE. Superficially, this is puzzling since SURF4 is important for the secretion of ApoB-containing lipoproteins (Saegusa et al., 2018; Wang et al., 2020). However, we found 10 CW motifs in ApoB (Data S1), suggesting this alternate mode of engagement with SURF4 might drive ApoB secretion. Overall, our bioinformatic analysis supported our hypothesis that engagement with SURF4 might predict short transit times within the ER, which is driven by specific protein properties.

## Discussion

Our dissection of SURF4 function in ER export has revealed some surprising features. First, SURF4 can bind cargo co-translationally, contrary to prevailing models that nascent proteins gain export competency only after folding. We note that both cargoes for which we demonstrate co-translational SURF4 interaction oligomerize in the presence of Ca²⁺. High affinity binding to SURF4 for such proteins has been proposed to drive rapid export, thereby limiting premature oligomerization within the Ca²⁺-rich ER (Yin et al., 2018). Indeed, mutations in ER-ESCAPE motifs of dentin proteins result in ER retention, with lumenal oligomerization proposed to be a cause of dentinogenesis imperfecta (von Marschall et al., 2012). Our unbiased analysis of ER-ESCAPE signals suggests Ca²⁺-binding proteins broadly have a high affinity for SURF4 and thus undergo rapid ER export. Retention of non-oligomeric Ca²⁺-binding proteins might sequester lumenal Ca²⁺. We note that deletion of SURF4 does not induce ER stress (Huang et al., 2021),

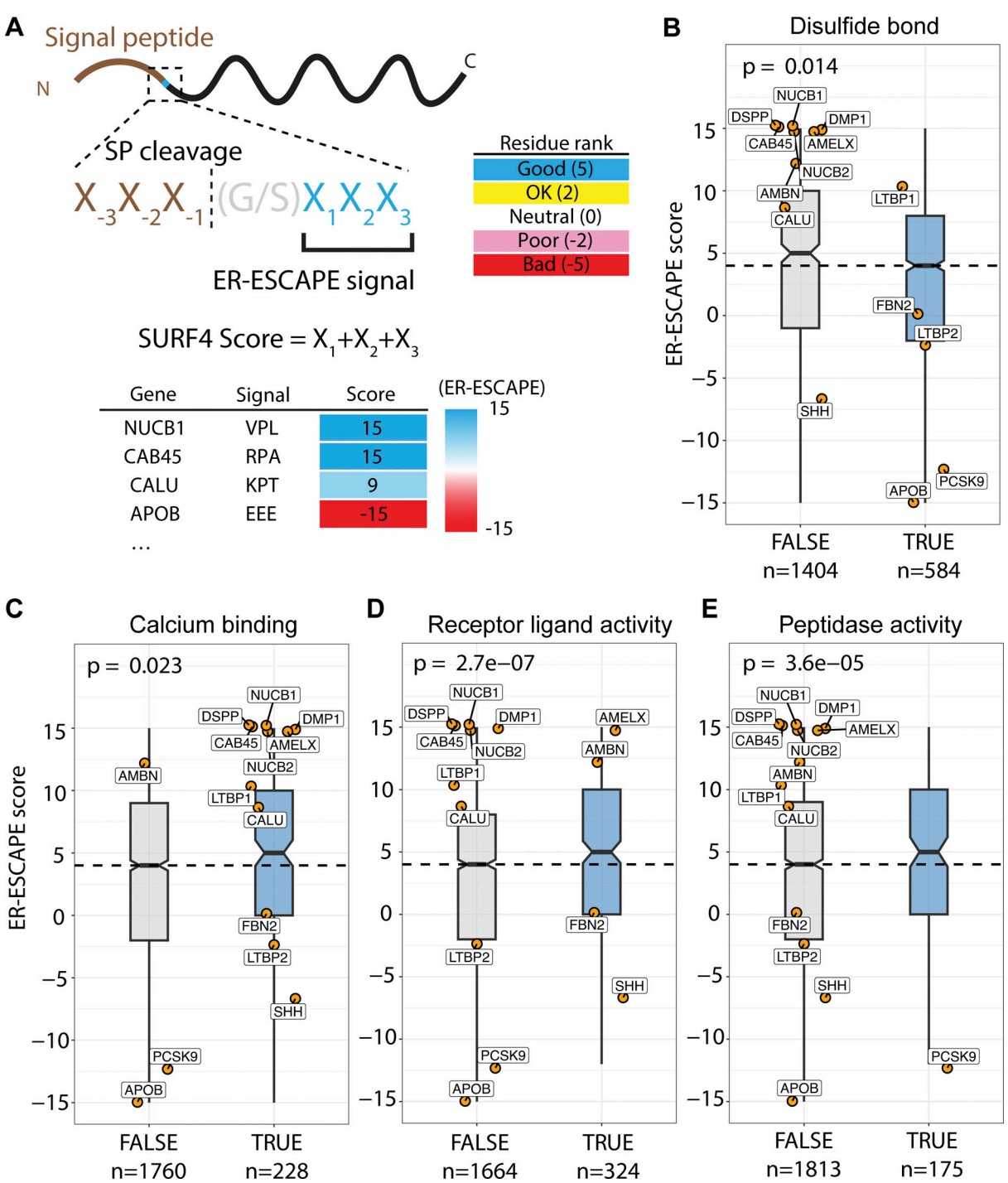

Figure 6. **ER-ESCAPE strength correlates with cargo properties. (A)** Schematic illustrating calculation of ER-ESCAPE score. Details on position-specific scoring matrix are included in Fig. S5 A. **(B–E)** Distribution of ER-ESCAPE scores of cargoes among different classes. Cargoes with (B) disulfide bond annotation have lower ER-ESCAPE scores, whereas cargoes with (C) calcium binding, (D) receptor ligand activity, and (E) peptidase functions have significantly higher ER-ESCAPE scores than the population median. In each plot, dashed lines represent median ER-ESCAPE score of the full dataset ($n = 1988$), boxes represent inter-quartile ranges, whiskers represent the range of distribution, notches represent 95% confidence interval of the median; curated SURF4 cargoes are highlighted as orange dots with gene names. Kruskal–Wallis test is used in each plot to test the significance and calculate a P value. The sample size is annotated in each plot below class labels.

suggesting our KO cells do not necessarily accumulate large amounts of aggregated protein and may have adapted to any changes to $Ca^{2+}$ homeostasis and flux.

The other major class of proteins with high apparent SURF4 affinity included proteases and cytokines/chemokines. Rapid ER export of proteases may be a safety mechanism that prevents

premature peptidase activity in the ER lumen, where vulnerable unfolded proteins might be destroyed. Such a mechanism would augment orthogonal safety features like low pH activation for lysosomal proteases. Additionally, co-translational binding to SURF4 might prevent certain classes of proteins from obtaining their correct fold in the ER, further providing protection against

premature activity or ligand binding. Why chemokines/cytokines and similar extracellular ligands might also undergo rapid export is less clear. We speculate that the folding needs of these proteins may be such that ER chaperones are not required, and removing these proteins from the folding milieu frees up the folding capacity for other clients. As proposed by Yin et al. (2018), perhaps these classes of proteins lack chaperone-binding sites and thus rely on co-translational SURF4 engagement to successfully ratchet through the Sec61 translocation channel (Gemmer and Förster, 2020; Matlack et al., 1999). Conversely, proteins with complex folding needs (e.g., disulfide bonds) seem to have lower SURF4 affinity, consistent with longer residence time in the ER. Correlating ER-ESCAPE strength with other features of secreted proteins should shed light on such models.

Our mutagenesis approach identified a region on SURF4 required for ER-ESCAPE binding. This lumenal pocket is lined with hydrophobic and negatively charged residues with the geometry and charge landscape suitable for binding positively charged and hydrophobic ER-ESCAPE motifs. It is possible that the presence of a neighboring N-glycan or engagement with the glycosylation machinery reduces binding due to steric effects. The position of glycosylation sites might thus be a rheostat that tempers ER-ESCAPE affinity. It remains to be elucidated how SURF4 releases its clients in the Golgi. A previous study proposed that CW motif binding to the SURF4 luminal helix in the ER is outcompeted by proteoglycans in the Golgi apparatus, thus releasing the cargo for onward trafficking and allowing SURF4 to be recycled (Tang et al., 2022a). By analogy with other conditional cargo-binding pathways (Wilson et al., 1993; Bräuer et al., 2019; Appenzeller-Herzog et al., 2004; Fujita et al., 2011; Muñiz and Riezman, 2016), ER-ESCAPE motif release from SURF4 may be driven by the drop in pH in the Golgi and/or SURF4 conformational changes that alter the geometry of the pocket. We note that the Alphafold prediction positions two histidine residues in proximity to the binding pocket, which might be a pH-responsive mechanism for altering cargo affinity. The interplay between ER-ESCAPE and CW motifs in driving cargo export is also an important factor to consider. Our bioinformatic analysis identified 747 proteins with at least one CW motif (Data S1), 101 of which also have a strong ER-ESCAPE score (12 or 15). Most interestingly, ApoB, which has an ER-ESCAPE score of –15, contains 10 predicted CW motifs. The mechanism, stoichiometry, and timing of how SURF4 binds such clients will be important to determine.

The second surprising feature of SURF4-mediated export is that distinct cargo clients appear to load into COPII vesicles via different mechanisms. ER export of PCSK9 is highly dependent on SEC24A (Chen et al., 2013; Gomez-Navarro et al., 2022), whereas Cab45 and NUCB1 require SEC24C/D. The full mechanism by which lumenal cargo dictates cytoplasmic adaptor binding remains to be determined. Nonetheless, our mutagenesis studies suggest a plausible model that builds on the concept of distinct cargoes having different transit times to exit the ER. PCSK9 becomes export competent upon self-cleavage of its propeptide, suggesting significant ER resident time to acquire folding and catalytic activity. Pro-PCSK9 would interact first with TMED10, likely as part of a homo- or hetero-oligomeric p24 family complex.

Once the PCSK9 ER-ESCAPE signal is revealed, SURF4 will bind and the dual-receptor complex will simultaneously bind two sites on SEC24A (D-site::SURF4; B-site::TMED10) for recruitment into vesicles. In contrast, Cab45 and NUCB1 seem to have a more conventional path to COPII vesicles, whereby C-terminal motifs in SURF4 engage the B-site on SEC24C/D to drive export. A possible explanation for why SURF4 C-terminal motif mutations did not affect SEC24A binding to SURF4 is that the TMED10 cytosolic tail likely contains higher affinity motifs than those of SURF4, thus outcompeting it at the B-site.

Our mechanistic studies of SURF4-mediated export from the ER show surprising diversity in how this single receptor drives the export of its clients. SURF4 possesses multiple lumenal sites of cargo interaction, coupled with multiple cytoplasmic ER export signals that engage multiple SEC24 paralogs via distinct sites. Moreover, by engaging clients co-translationally, SURF4 may also have some chaperone/holdase function. In this light, it is not surprising that a sizeable number of secreted proteins might be expected to engage with SURF4. How many of these also require co-receptors, like the PCSK9/TMED10 example, remains to be seen. A better understanding of the biophysical properties of SURF4 clients and their mode of conditional binding will further enhance our understanding of this conserved driver of protein secretion.

## Materials and methods

### Cell lines and cell culture
HEK-293TREx, HEK-293T, and Huh7 cell lines used in this study (Gomez-Navarro et al., 2022) were cultured in Dulbecco's Modified Eagle's Medium (DMEM; Gibco) with 10% fetal bovine serum (Qualified FBS; Gibco) at 37°C and 5% $CO_2$. HEK-293TREx SURF4 KO, SEC24A KO, SEC24B KO, Huh7 SURF4 KO, SEC24C KO, SEC24D KO, and HEK-293TREx TMED10 KO were previously described (Gomez-Navarro et al., 2022; Huang et al., 2021; Zavodszky and Hegde, 2019). HEK-293T double SURF4 and SEC23C KO was generated by knocking out SURF4 on top of SEC23C using the same guide for SURF4 locus as for a single mutant (5′-GGCCCAGAACGGAGCCGCCT-3′) (Huang et al., 2021). Multiple single-cell–derived KO clones were isolated and screened for gene disruption by amplifying and sequencing the edited genomic site and western blotting. A cell line stably expressing FLAG-SURF4 was generated by transfecting HEK-293TREx SURF4 KO with pcDNA5/FRT/TO-FLAG-SURF4 and pOG44 (encoding Flp-Recombinase) at 1:9 ratio; integrants were selected using 7.2 µg/ml Blastitcidin S and 1.3 µg/ml Hygromycin B by exchanging media every 2–3 days in a 10-cm dish until confluent colonies were seen. FLAG-SURF4 expression was tested by western blot (Fig. S4 A). Plasmid transfections were performed with OptiMEM (Gibco) and TransIT-293 (Mirus) for HEK cells, incubating for 16–24 h. All cell lines were routinely checked for mycoplasma contamination and tested negative.

### Antibodies
Anti-HA.11 (mouse, 901515; BioLegend) and anti-V5 (mouse, R960-25; Invitrogen) were used for pulse-chase IP, the latter was also used for DSP crosslinking IP. Anti-TRAPα (rabbit; Hegde Lab,

[Fons et al., 2003]) was used for in vitro crosslinking IP. The following were used for western blot at indicated dilutions: anti-βactin (mouse, 1:10,000, A1978; Sigma-Aldrich), Cab45 (rabbit, 1:2,000; von Blume lab), anti-SEC24A (rabbit, 1:1,000, 9678S; Cell Signalling), anti-SEC24C (rabbit, 1:1,000, 14676S; Cell Signalling), anti-LgBiT (mouse, 1:500, N7100; Promega, WB), anti-V5-HRP (1:10,000, ab1325; Abcam), anti-TMED10 cytosolic tail (rabbit, 1:1,000, BSYN6430; Hegde Lab) (Zavodszky and Hegde, 2019), Fig. 3 E, anti-TMED10 (rabbit, 1:2,000, A305-228A; Bethyl), Fig. S3 B, anti-TMED2 (rabbit, A305-467A; Bethyl [1:2,000], and anti-SURF4 [rabbit, 1:1,000; Chen Lab] [Wang et al., 2020], Fig. S4 A). Secondary antibodies were used at 1:10,000 dilution: Pierce anti-mouse from Thermo Fisher Scientific (31410), anti-rabbit from Sigma-Aldrich (A4914). The following were used for immuno-fluorescence at indicated dilutions: primary anti-FLAG (rabbit, 1:500, F7425; Sigma-Aldrich) and anti-SEC31A (mouse, 1:100, 612350; BD Biosciences) with secondary anti-mouse AlexaFluor-647 and anti-rabbit AlexaFluor-555 (1:250, A21235 and A21428; Thermo Fisher Scientific, respectively).

### Radiolabeled pulse-chase
As published in Gomez-Navarro et al. (2022), HEK-293TREx cells were grown in 10-cm dishes until they reached ~70% confluency and were transfected with the protein to be investigated. The next day, cells were washed with warm PBS, collected into 1.5-ml Eppendorf tubes, and incubated in starvation medium (DMEM lacking Met/Cys [Gibco] +10% FBS) at 37°C and 5% $CO_2$ for 30 min. Starvation medium was then replaced with pulse medium (starvation medium supplemented with 16 µCi/sample EasyTag EXPRESS $^{35}$S Protein Labeling Mix [PerkinElmer] or $^{35}$S-Met-label, [SCIS-103; Hartmann Analytic]) and cells were incubated for 30 min. Cells were then incubated with chase medium (complete DMEM +10% FBS) for the indicated time points after which they were treated with N-Ethylmaleimide (Sigma-Aldrich). Media was collected and cells were lysed in 50 mM Tris pH 7.4, 1% Triton X-100, 150 mM NaCl, 2 mM EDTA, 0.5 mM PMSF, and protease inhibitor (Roche) on ice for 10 min. Lysates, cleared by centrifugation, and media were precleared with Protein-G Sepharose beads 4 Fast Flow (GE Healthcare) and the protein of interest was immunoprecipitated using Protein-G Sepharose beads coupled with monoclonal mouse anti-HA.11 antibody or anti-V5. Radiolabeled immunoprecipitated proteins were eluted in sample buffer (50 mM Tris, pH 6.8, 0.1% [vol/vol] glycerol, 20% [p/vol] SDS, 5% [vol/vol] β-mercaptoethanol, 1 mg/ml Bromophenol Blue), separated on NuPAGE 4–12% Bis-Tris gels (Thermo Fisher Scientific), and detected by phosphor imaging using a Typhoon scanner (GE Healthcare). Protein bands were quantified using Fiji by measuring the mean grey value and subtracting the background of the same size ROI from each protein lane. The percentage of secreted or mature band at each time point was quantified by dividing the mean grey value by [t = 0'secreted + intracellular] mean grey value. Results were plotted and statistically analyzed in Prism 8 or 9 (GraphPad Software).

### Cloning
NUCB1-HA, Cab45-HA, FLAG-SURF4 cloned in pcDNA3.1, Cab45-HA cloned in pLPCX, and PCSK9-V5 cloned in pCDNA5/

FRT/TO were described previously (Gomez-Navarro et al., 2022). For in vitro transcription of NUCB1-HA and Cab45-HA, the ORFs were cloned into a plasmid carrying an SP6 promoter (Hegde et al., 1998) using BglII site at the 5′ end and EcoRI site at the 3′ end. FLAG-SURF4 ORF was subcloned into pcDNA5/FRT/TO vector under tet promotor using BamHI site at 5′ end and NotI site at the 3′ end. SEC24C ORF from pcDNA3.1-N-term-Myc-6xHis (Bisnett et al., 2021) was subcloned into pFN35K SmBiT TK-neo Flexi Vector (Promega) using SgfI site at the 5′ end and NotI site at the 3′ end. Site-directed mutagenesis was carried out using QuikChange Lightning (one mutational site at a time using forward and reverse primers) or QuikChange Lightning Multi (one or more mutational sites, using forward-only primers) kits (Agilent) according to the manufacturer's instructions. Oligonucleotide primer sequences are listed in Table S1.

### NanoBiT luciferase complementation assay
For NanoBiT measurements, HEK-293TREx double KO (SEC24A or SEC23C KO with SURF4 KO) cells were plated at a density of $8 \times 10^3$ cells/well in a clear-bottom white-wall 96-well plate (GreinerBio) precoated with D-poly-lysine (Gibco), cultured in DMEM +10% FBS, and transfected the following day with the corresponding NanoBiT constructs. NanoGlo Live Cell Substrate was diluted 1:5 in LCS buffer (Promega) and further 1:20 in OptiMEM, added to cells, and incubated at room temperature for 15 min. For 4-PBA experiments, cells were treated with indicated concentrations for 4 h prior to adding NanoGlo. Luminescence was measured using a Spark10M plate Reader (Tecan).

For investigating the effect of SURF4 mutants, cells were plated on poly-lysine (Gibco) coated plates. 24 h after transfection, cells were washed once with warm PBS just before applying NanoGlo.

### Western blot
For analysis of total cellular proteins, cells were lysed in 1%SDS-100 mM Tris pH 8.0 and supplemented with protease inhibitor. Cells in lysis buffer were heated at 95°C for 10 min, shaking at 800 rpm with small glass beads. Protein concentrations were adjusted based on A280 values to a final concentration of 2 µg/µl in sample buffer and heated at 55°C for 10 min or 95°C for 2 min; 30 µg of proteins were then separated on NuPAGE 4–12% Bis-Tris gels (Thermo Fisher Scientific). When collecting condi-tioned media samples for immunoblotting, complete DMEM was exchanged to OptiMEM for 16 h. 1.5 ml of media was collected in Eppendorf tubes, protease inhibitors were added, and samples were spun at 4°C 1,000 rpm for 5 min to remove the remaining cells. 1 ml of media was then concentrated to 50 µl using Amicon Ultra-4 filter units at 4°C (Milipore) and heated for 10 min at 55°C with 3x sample buffer, of which 15 µl was loaded onto NuPAGE 4–12% Bis-Tris gels. Proteins were transferred to methanol-activated 0.45 µM pore PVDF membrane (Immobilon-P; Merck) and then blocked in 5% fat-free milk dissolved in TBST (0.1% Tween20 [Sigma-Aldrich] in TBS) for 1 h at room tem-perature (RT). Incubation with primary antibodies was done overnight at 4°C and with secondary 1 h at RT in 5% milk-TBST, washing 3 × 5 min in TBST after each incubation. Protein bands

were visualized using HRP-conjugated secondary antibodies and chemiluminescent substrate (Immobilon Western Chemiluminescent HRP Substrate [Merck or Cytiva] for weak antibodies or ECL western blotting Reagent [Merck] at 1:1 ratio for stronger antibodies) and detected with an X-ray film.

## Protein structure prediction, conservation, and disease variants

The initial SURF4 structure used for generating the SURF4 mutant library was predicted using trRosetta (Yang et al., 2020) default settings. Other servers used for TMD predictions were TOPCONS (Tsirigos et al., 2015), OCTOPUS, Philius, Poly-Phobius, SCAMPI, SPOCTOPUS as well as SURF4 sequence alignment with known Erv29 TMDs (Foley et al., 2007). Conservation was predicted using ConSurf (Ashkenazy et al., 2016) default settings. Further SURF4 structure prediction and co-folding were performed using AlphaFold2 and Alpha-Fold2 Multimer installed on LMB's computer cluster. gnomAD v2 was used to look at missense disease variants. Only those missense mutations were kept that had a score of allele count >1. gnomAD v3 was used to annotate LoF mutations (Karczewski et al., 2020).

## Immunofluorescence microscopy

Huh7 SURF4 KO cells were seeded on coverslips in six-well plates and transfected the next day at ~50% confluency with the indicated constructs. 24 h later, cells were fixed with 4% paraformaldehyde for 15 min, permeabilized with 0.1% Saponin for 10 min, blocked with 1% bovine serum albumin for 1 h, and stained with primary antibodies for 1.5 h at RT in a humidity chamber; staining with secondary antibodies followed for 1 h at RT in a humidity chamber. Last, cells were stained with 10 µg/µl DAPI (Merck) for 15 min. Coverslips were washed five times in PBS after fixing, primary and secondary antibody incubations, and DAPI staining. Coverslips were mounted in Prolong Diamond Antifade (Thermo Fisher Scientific). Z-stack images at 0.9-µm intervals were taken on a Zeiss LSM 710 confocal microscope with 63×/1.4NA oil immersion objective at room temperature. Images were collected using ZEN software, followed by processing in Fiji-2, where maximum intensity Z-projections were used, and brightness and intensity of each channel were manually adjusted.

## Knock-downs

For siRNA-mediated knock-downs, cells were seeded to be ~30–40% confluent at the time of transfection. After seeding, cells were treated with the indicated siRNA the same day by first diluting Silencer Select siRNA (Invitrogen) in OptiMEM and separately combining Lipofectamine RNAiMax with OptiMEM in aliquots of the same volume (1:34 RNAiMax:OptiMEM ratio). Then siRNA-OptiMEM was added into RNAiMax-OptiMEM, incubated for 20 min, and applied to cells drop-wise to have siRNA at a final 20 nM concentration. The following Silencer Select siRNAs (Thermo Fischer Scientific) were used in this study: TMED2-1 (s21570), TMED2-2 (s21571), TMED10-1 (s21600), TMED10-2 (s21601), SEC24A (s21226), SEC24C (s18516), negative control Nr. 1 cat. no. 4404021. Cells were

treated for 1–3 days as indicated and for 2 days in pulse-chase assays.

## DSP crosslinking co-immunoprecipitation

HEK TREx-293 TMED10 KO cells were seeded in 6 well-plates and co-transfected with the indicated plasmids the next day at 70% confluency. One day later, cells were collected in PBS and incubated in PBS supplemented with 0.2 mM CaCl$_2$ and 0.2 mM DSP (22585; Thermo Fisher Scientific), rotating end-over-end at RT for 30 min. The crosslinking reaction was quenched by adding 25 mM Tris-HCl and incubating cells on ice for 15 min. Cells were lysed in 200 µl IP lysis buffer (50 mM Tris-HCl, 150 mM NaCl, 2 mM CaCl$_2$, and 1.0% Triton X-100, supplemented with protease inhibitors, pH 7.5). Cleared lysates were incubated with anti-HA agarose (A2095-1ML; Sigma-Aldrich), preincubated for 1 h at 4°C with IP Blocking Buffer (50 mM Tris-HCl, 500 mM NaCl, 2 mM CaCl$_2$, 5% BSA, pH 7.5) to reduce non-specific binding, or Protein G Sepharose (GE Healthcare) preincubated with anti-V5. Beads were washed with IP Blocking Buffer two times and three times with IP lysis buffer prior to a 30 min elution at 55°C with 3x sample buffer.

## Preparation of semi-permeabilized cells

80–90% confluent HEK-293TREx cells were collected in ice-cold PBS and washed twice. In the case of FLAG-SURF4 cell lines, they were induced with 0.2 µg/ml doxycycline for 3 h prior to collection. Cell mass was weighed in preweight Eppendorf tubes. Cells were permeabilized on ice for 10 min in 1xPSB buffer (50 mM HEPES pH 7.4, 100 mM KOAc, 2.5 mM Mg[OAc]$_2$) containing 0.01% purified digitonin. Cells were then collected by centrifugation using a short (15 s) spin at 4°C, washed twice in 1xPSB, and finally resuspended in 0.5xPSB depending on their mass (10 mg of non-permeabilized cell mass was resuspended in 23 µl 0.5xPSB). Where indicated, before resuspending in 0.5xPSB, semi-permeabilized cells were treated with 150 U/ml S7 nuclease (Roche) in 1xRNC supplemented with 1 mM CaCl$_2$ to reduce endogenous mRNA. Nuclease digestion was performed for 10 min on ice, quenched by adding 2 mM EGTA, and cells were collected by centrifugation and washed twice with 1xRNC. Semi-permeabilized cells were used immediately for IVT.

## In vitro transcription, translation, and site-specific photo-crosslinking

In vitro transcription and translation were based on a previously described method by Sharma et al. (2010). In short, mRNAs to be translated were in vitro transcribed for 1 h at 37°C using 100 µg of PCR product per 10 µl reaction in homemade cT1 mix (40 mM HEPES pH 7.4, 6 mM MgCl$_2$, 20 mM spermidine, 10 mM reduced glutathione, 0.5 mM ATP, 0.5 mM UTP, 0.5 mM CTP, 0.1 mM GTP, 0.5 mM CAP), 0.4–0.8 U/ml RNasin (Promega), and 0.4 U/ml SP6 polymerase (NEB Biolabs).

IVT was run for 10 min per 10 kDa of translated protein at 32°C. IVT reactions contained 35% volume of nuclease-treated rabbit reticulocyte lysate (Green Hectares), 20 mM HEPES, 10 mM KOH, 40 µg/ml creatine kinase, 50 µg/ml tRNA purified from pig liver, 12 mM creatine phosphate, 1 mM ATP, 1 mM GTP, 50 mM KOAc, 2 mM MgCl$_2$, 1 mM reduced glutathione, 0.3 mM

spermidine, and 40 μM of each amino acid except methionine (all of the above contained in the homemade cT2 mix), with radiolabeled methionine added separately at 0.5 μCi/μl (EasyTag L-[³⁵S]-Methionine; PerkinElmer). IVT reactions were then further supplemented with 10% vol of semi-permeabilized cells and 5% vol of the transcript taken directly from in vitro transcription reaction. Where indicated, cavinafungin was added last to the IVT master mix and incubated on ice for 10 min.

For site-specific photo-crosslinking, IVT reactions were supplemented with 0.1 mM Bpa, 5 μM *B. Stearothermophilus* transfer RNA (tyrosyl-tRNA) with a CUA anti-codon and 0.25 μM BPA-tRNA synthetase for Bpa to be incorporated using amber codon suppression at desired positions as previously described in Lin et al. (2020). After translation was completed, reactions were layered on a 20% sucrose cushion in 1xPSB and spun at 4°C in a microcentrifuge for 5 min at 10,000 rpm. The resulting membrane pellet was resuspended in 1xRNC and irradiated under UV (254 nm) for 10 min, 1.5 cm away from the light source, on an ice block. To recover a clear covalent crosslink, proteins of interest were immunoprecipitated under denaturing conditions. Cells were lysed in 1%SDS-100 mM Tris by heating at 95°C and vortexing until DNA was sheared, cleared lysates split in half and diluted 1:15 in denaturing IP buffer (50 mM HEPES, 200 mM NaCl, 1% Triton X-100) containing either washed HA-agarose (A2095-1ML; Sigma-Aldrich) or anti-FLAG M2 affinity gel (A2220; Sigma-Aldrich), or Protein G or A Sepharose with a corresponding antibody. 30 μl IVT reaction was used for a single protein IP. Samples were immunoprecipitated at 4°C for 1.5 h rotating end-over-end. Beads were washed three times with IP buffer and samples were eluted in 3x sample buffer by heating at 95°C for 2 min. Proteins were separated on 4–12% Bis-Tris gels (Invitrogen), which were then fixed, dried, and exposed to either phosphorscreen or X-ray film.

### Proteinase K (PK) protection assay

IVT reactions were carried out as described in the above section. Once the reaction was complete, 1 μl was diluted into 19 μl 3x sample buffer and heated at 95°C (-PK treatment). To the remaining 19 μl, 0.5 mg/ml PK (Thermo Fisher Scientific) was added. The reaction was incubated on ice for 1 h. before 250 U/ml of benzonase was added, and the sample was incubated on ice for a further 10 min to digest the released DNA. Reactions were quenched with 6.25 mM fresh PMSF for 3 min on ice. PK-digested samples were diluted 1:10 in 1% SDS-100 mM Tris, heated at 95°C, and vortexed interchangeably for 3 min, then digested samples were diluted 2:1 in 3x sample buffer. 10 μl of each sample was loaded.

### Bioinformatics

Curated human proteome entries were acquired from UniProt (The UniProt Consortium, 2023) and filtered as follows to generate a putative soluble secretome for analysis: (1) proteins with annotated transmembrane domain were excluded; (2) proteins without annotated SP were excluded; (3) proteins with annotated glycosylphosphatidylinositol (GPI)-anchor were excluded. In the next step, two scores were calculated for each protein based on the scoring matrix from Fig. S5 A based on previously reported ER-ESCAPE motif consensus (Yin et al., 2018): (1) a score for residues at positions +1, +2, +3 after predicted SP cleavage site, termed the tripeptide score; (2) a score for residues +2, +3, +4 positions according to the same matrix, but only when +1 position is glycine or serine, termed the tetrapeptide score. The higher of the two was assigned as the ER-ESCAPE score for each motif. This scoring pipeline was devised to accommodate possible SURF4 substrates with a small amino acid at +1 position (Yin et al., 2018; Gomez-Navarro et al., 2022). Sequence manipulations and score calculations were performed using the BioPython package (Cock et al., 2009). This putative soluble secretome annotated with ER-ESCAPE scores is included in Data S1.

The soluble secretome was then assessed for ER-ESCAPE score based on various functional or structural annotations in UniProt. For the classes reported in Fig. 6 and Fig. S5,

(1) Calcium binding (Fig. 6 C) refers to entries in the union of three positive criteria:
  (a) Annotation of calcium-binding site "*CHEBI:29108*" in the section Binding Site;
  (b) Annotation of gene ontology (GO) term binding to calcium ion "*GO:0005509*" in section GO molecular function (MF);
  (c) Annotation of GO term hydroxyapatite binding "*GO:0046848*" in section GO MF.
(2) EF hand (Fig. S5 B) refers to entries with annotation under rules: *EF_HAND1 (PS00018)* and *EF_HAND2 (PS50222)* in section PROSITE.
(3) Multiple calcium binding (Fig. S5 C) refers to entries with at least two separate calcium binding sites annotated in the section Binding Site.
(4) Receptor ligand activity (Fig. 6D) refers to entries in the union of two positive criteria:
  (a) Annotation of GO term receptor ligand activity "*GO:0048018*" in section GO molecular function (MF);
  (b) Annotation of any other GO term that is a child term of "*GO: 0048018.*" This includes: hormone activity "*GO:0005179,*" death receptor agonist activity "*GO:0038177,*" pathogen-derived receptor ligand activity "*GO:0140295,*" chemoattractant activity "*GO:0042056,*" neuropeptide activity "*GO:0160041,*" morphogen activity "*GO:0016015,*" receptor ligand inhibitor activity "*GO:0141069,*" growth factor activity "*GO:0008083,*" cytokine activity "*GO:0005125,*" pheromone activity "*GO:0005186,*" chemorepellent activity "*GO:0045499,*" opioid peptide activity "*GO: 0001515.*"
(5) Peptidase activity (Fig. 6 E) refers to any entry with annotated Enzyme Commission number (EC number) in class peptidase "3.4.*.* ."
(6) Disulfide (Fig. 6 B) refers to any entry with annotations in the section Disulfide Bond.

GO enrichment analysis was performed using PANTHER (Mi et al., 2019; Thomas et al., 2022) for two subgroups of cargoes: (1) cargoes with ER-ESCAPE score of 15 and 12, named "Positive ER-ESCAPE Score" (Fig. S5 D); (2) cargoes with ER-ESCAPE score of –15 and –12, named "Negative ER-ESCAPE Score," (Fig. S5 E) using "*GO molecular function*" option, and whole human genome

as a reference list. The results were included in Data S1. The resulting enriched list was further filtered with positive enrichment and with the number of observations being >5% of the total list, to create a confident curated list, before being plotted in R using *ggplot2* package (Wickham, 2016).

### Online supplemental material

Fig. S1 shows Cab45 secretion in Sec24B, SEC24C, and SEC24D single KO lines; stability of SEC24A, SEC24C, and SURF4 mutants used in the NanoBiT assay; a topology diagram of SURF4 and associated point mutants; and immunoblots showing stability of SURF4 mutants and Cab45 secretion phenotypes. Fig. S2 shows the subcellular localization of SURF4 and various point mutants. Fig. S3 shows the identification and characterization of TMED10 and TMED2 as candidate co-receptors selective for PCSK9 export. Fig. S4 shows a series of experiments that establish the in vitro translation system for Cab45 and NUCB1 in SURF4 stably transfected cell lines, including analyses of adducts created by amber stop incorporation at different sites, dependence on ER-ESCAPE motifs and SP cleavage. Fig. S5 shows details of how we generated the ER-ESCAPE score along with analyses of ER-ESCAPE strength for calcium-binding proteins, and GO enrichment for strong and weak ER-ESCAPE-containing proteins. Table S1 contains three sheets: a summary of SURF4 mutants and their phenotypes; oligonucleotide sequences used for site-directed mutagenesis and cloning; and information on commercial siRNA reagents used in knockdown experiments. Data S1 contains information used in the calculation and analysis of ER-ESCAPE scores presented in Figs. 6 and S5.

### Data availability

All data are included in the article or supplementary data. Materials are available on request from the corresponding author.

## Acknowledgments

We thank Ramanujan Hegde (MRC LMB) for critical feedback on the manuscript and his lab members for guidance with in vitro translation assays, in particular, Szymon Juszkiewicz for help in establishing the assay, and Eszter Zavodszky for TMED10 reagents. We thank Julia von Blume (Yale University) and Xiao-Wei Chen (Peking University) for Cab45 and SURF4 antibodies, respectively, and Nik Sergejevs and Iqball Duloo (University of Oxford) and Martin Spiess (University of Basel) for providing cavinafungin.

This work was supported by the Wellcome Trust (225216/Z/22/Z) and the Medical Research Council as part of United Kingdom Research and Innovation (also known as UK Research and Innovation) (MC_UP_1201/10). For the purpose of open access, the MRC Laboratory of Molecular Biology has applied a CC BY public copyright license to any author accepted manuscript version arising. Open Access funding provided by the University of Dundee.

Author contributions: J. Maldutyte: Conceptualization, Formal analysis, Investigation, Methodology, Validation, Visualization, Writing - original draft, Writing - review & editing, X.-H. Li: Data curation, Formal analysis, Investigation, Methodology, Visualization, Writing - review & editing, N. Gomez-Navarro: Investigation, Resources, E.G. Robertson: Resources, E.A. Miller: Conceptualization, Formal analysis, Funding acquisition, Project administration, Supervision, Visualization, Writing - original draft, Writing - review & editing.

Disclosures: The authors declare no competing interests exist.

Submitted: 17 June 2024

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

**Supplemental material**

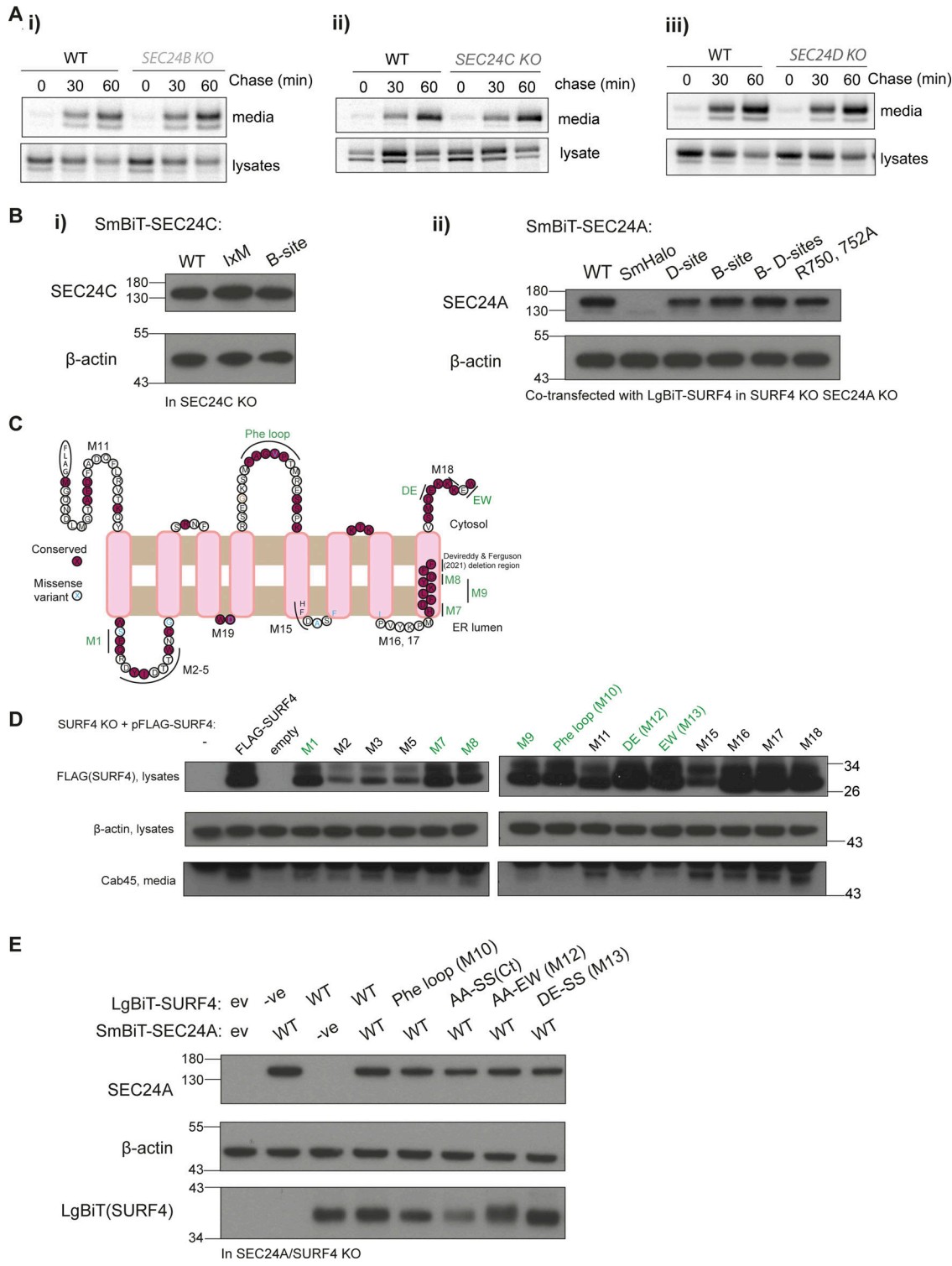

**Figure S1. Cargo secretion in SEC24C KO, NanoBiT mutant stability and SURF4 mutagenesis. (A)** Radiolabeled pulse-chase experiment testing Cab45 secretion in WT and SEC24 KO cells. Transiently transfected HA-tagged Cab45 was immunoprecipitated from media and lysates at indicated time points after [$^{35}$S]-Met/Cys addition and detected by SDS-PAGE and phosphorimage analysis. $n$ = 1. **(B)** Stability of indicated SEC24 constructs was tested in appropriate KO cells by immunoblotting for steady-state protein levels using the antibodies indicated. β-actin served as loading control. **(C)** SURF4 topology prediction based on trRosetta, TOPCONS, and other topology prediction algorithms (see Materials and methods), with conservation and disease variants mapped from gnomAD. Based on this prediction, mutants in cytosolic or lumenal regions were generated. Those labeled in green indicate defects in Cab45 secretion but stable expression of the mutant. **(D)** Immunoblots of steady-state FLAG-SURF4 and secreted Cab45 upon transient expression of FLAG-SURF4 WT and mutant constructs in HEK-293TREx SURF4 KO. Actin serves as a loading control for lysates. Mutants in green had reduced or no Cab45 secretion rescue in the media. **(E)** Stability of indicated SURF4 constructs was tested in appropriate KO cells by immunoblotting for steady-state protein levels using the antibodies indicated. β-actin served as a loading control. Source data are available for this figure: SourceData FS1.

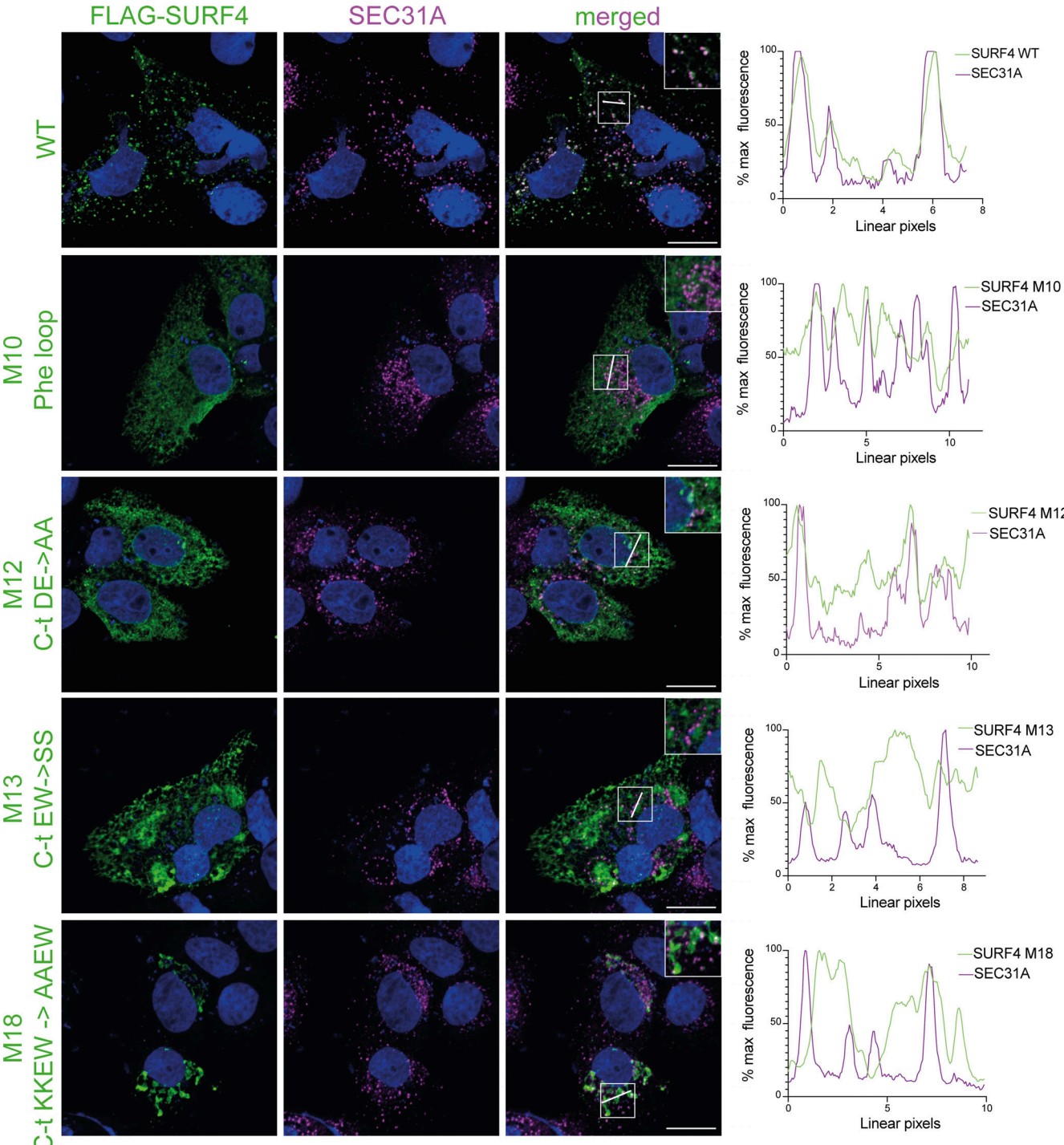

**Figure S2. FLAG-SURF4 WT and cytosolic mutant subcellular localization.** FLAG-SURF4 WT and indicated mutants were transiently transfected into Huh7 SURF4 KO cells. Cells were fixed, immunostained using anti-FLAG and anti-SEC31A antibodies, and imaged on a confocal microscope. Plots indicate FLAG-SURF4 and SEC31A co-localization along the indicated line in each instance. Scale bar = 15 μm. Inset diameters are 16 μm.

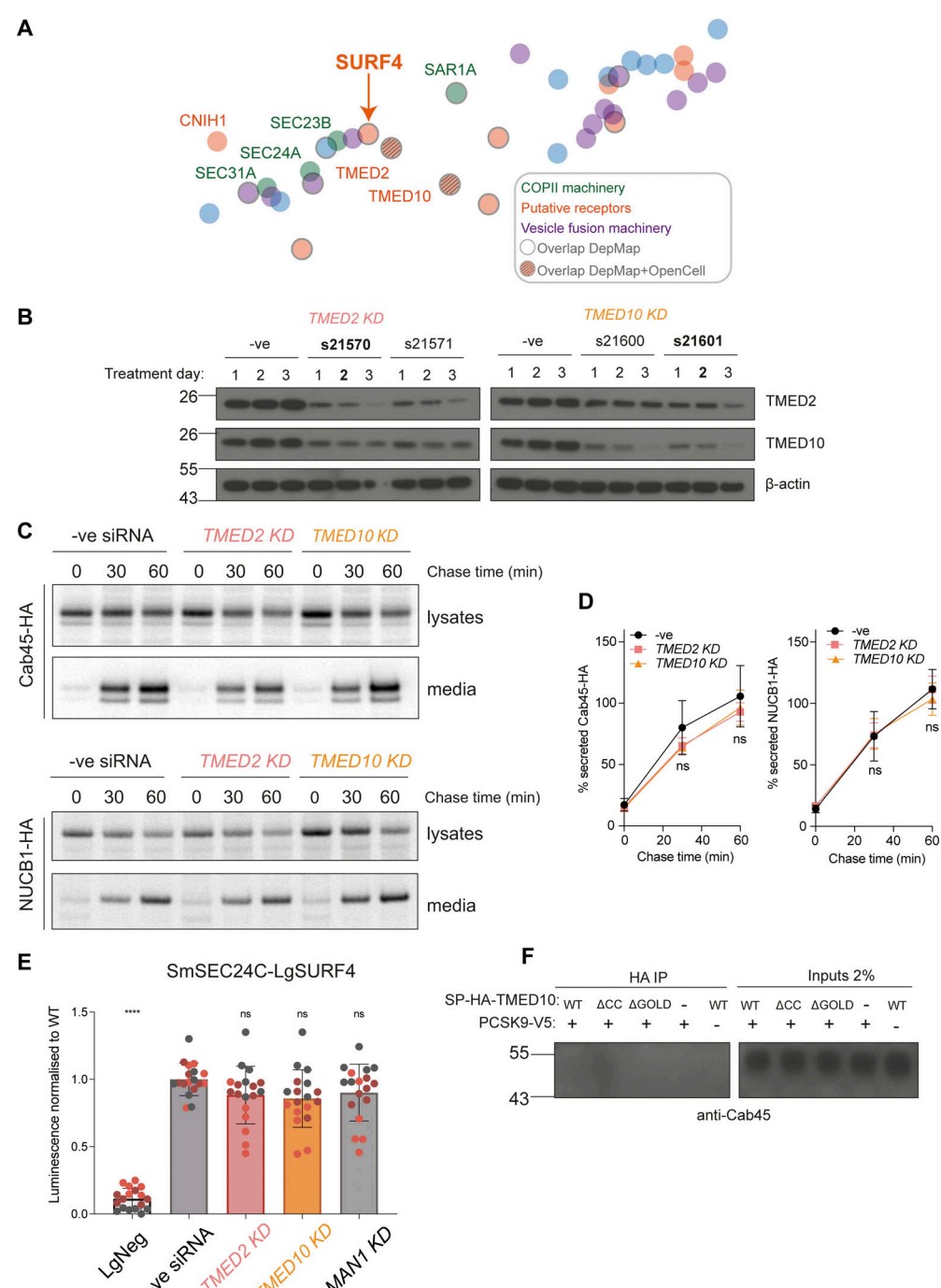

Figure S3. **TMED2 and TMED10 do not participate in Cab45 or NUCB1 secretion. (A)** Close-up view of the co-essentiality Browser's neighborhood "ER-to-Golgi body transport". Wainberg et al. (2021) visualized this co-essentiality by plotting strongly co-essential genes together, as determined by their generalized least squares method. Overlapping Depmap co-dependencies are outlined in grey and overlapping DepMap+OpenCell interactions have a grey grid added. **(B)** HEK-293TREx cells were treated with the indicated Silencer Select siRNAs for the indicated number of days, alongside a negative siRNA control. For each TMED treatment, lysates were blotted for both TMED2 and TMED10. Actin served as loading control. Conditions in bold were chosen for pulse-chase and NanoBiT experiments. **(C)** Radiolabeled pulse-chase of Cab45 and NUCB1. HEK-293TREx cells were transfected with the indicated siRNAs, then 24 h later transfected with a plasmid expressing Cab45-HA or NUCB1-HA, which were detected by pulse-chase and immunoprecipitation the following day. Protein secretion was quantified from autoradiographs following SDS-PAGE. Each pulse-chase experiment is representative of three biological replicates and is quantified in D. **(E)** SURF4/SEC24C double KO cells were transfected with the indicated siRNAs, then 24 h later were cotransfected with SmBiT-SEC24C and LgBiT-SURF4 NanoBiT constructs. Luciferase luminescence values were measured and normalized to WT. Triangles represent mean and error bars represent SD. **(F)** DSP-crosslinking co-immunoprecipitation of endogenous Cab45 from cells expressing SP-HA-TMED10 WT, coiled-coil, and GOLD domain deletion mutants. Statistical tests were one-way ANOVA with Dunnett's correction for multiple testing. Data distribution was assumed to be normal but this was not formally tested. ns = not significant, **** = P value <0.0001. For each NanoBiT experiment, six technical replicates were used in each of the three independent biological replicates, as indicated by differential coloring within superplots. Triangles represent the mean and error bars represent SD. Source data are available for this figure: SourceData FS3.

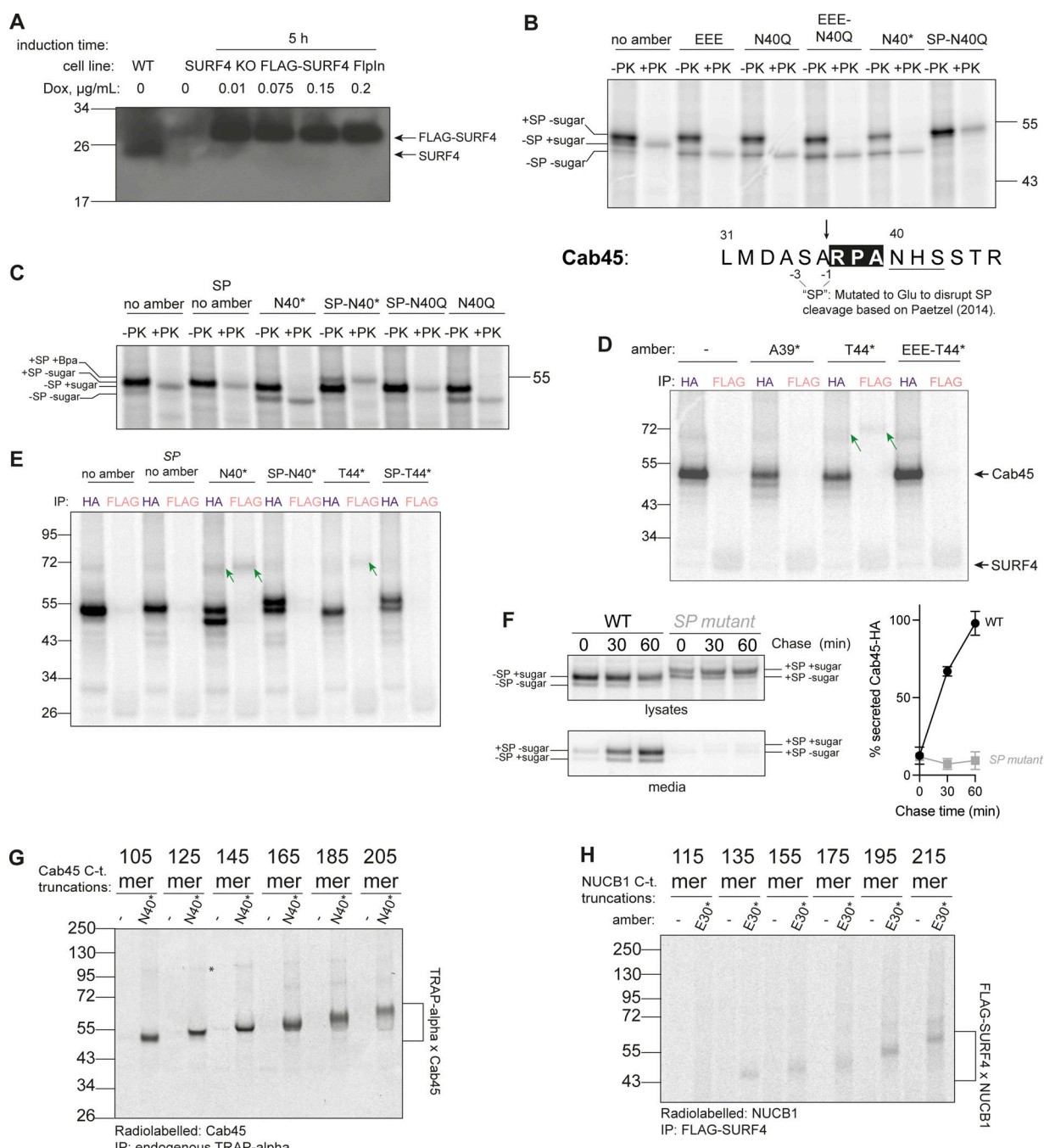

Figure S4.   **Co-translational and SP-dependent interaction of clients with SURF4. (A)** Doxycycline induction test to optimize concentration for FLAG-SURF4 expression. A western blot of cell lysates, probed with an antibody for endogenous SURF4 is shown. **(B)** Proteinase K (PK) protection assay for various Cab45 constructs, examining the effects of indicated mutations at and near the SP cleavage, ER-ESCAPE and *N*-glycosylation sites with respect to glycosylation and SP cleavage. The size of PK-protected (i.e., ER membrane-enclosed) Cab45-40N* band is the same as that of a glycosylation mutant Cab45-N40Q (and Cab45-EEE [ER-ESCAPE mutation] as well as Cab45-EEE-N40Q), lower than that of Cab45 WT (no amber) and much lower than Cab45-SP-N40Q, whereby SP cleavage is disrupted (as per schematic below). This indicates that Bpa incorporation at position 40 (as well as ER-ESCAPE mutation into EEE) does not disrupt SP cleavage but instead prevents *N*-glycosylation. The higher Mw form of Cab45-40N* seen in Fig. 4 B IPs is therefore likely SP-uncleaved form that remains in close proximity to the ER membrane and therefore is IPed together. The gel was ran until 26 kDa marker ran out to have better separation of the different Cab45 intermediates. **(C)** Proteinase K (PK) protection assay of various Cab45 constructs examining the effects of indicated mutations at and near the SP cleavage, ER-ESCAPE and *N*-glycosylation sites with respect to membrane glycosylation and SP cleavage. The gel was ran until 17 kDa marker ran out to have better separation of the different Cab45 intermediates. **(D)** Site-specific Cab45-HA photo-crosslinking experiment showing Cab45-SURF4 direct interaction is ER-ESCAPE-dependent and still occurs with the *N*-glycan present. Semi-permeabilized cells were treated with S7 nuclease to reduce doxycycline-induced FLAG-SURF4 background. **(E)** Samples as in C, but Cab45-HA and FLAG-SURF4 IPs were performed from UV-crosslinked semi-permeabilized cells. Green arrows indicate Cab45-SURF4 cross-links. **(F)** n = 2. **(G)** Cab45 C-terminal truncations photo-crosslinked to TRAPα with Bpa placed at the *N*-glycosylation site of Cab45 (N40*). **(H)** NUCB1 C-terminal truncations photo-crosslinked to FLAG-SURF4 immunoprecipitated from UV-crosslinked semi-permeabilized cells. Source data are available for this figure: SourceData FS4.

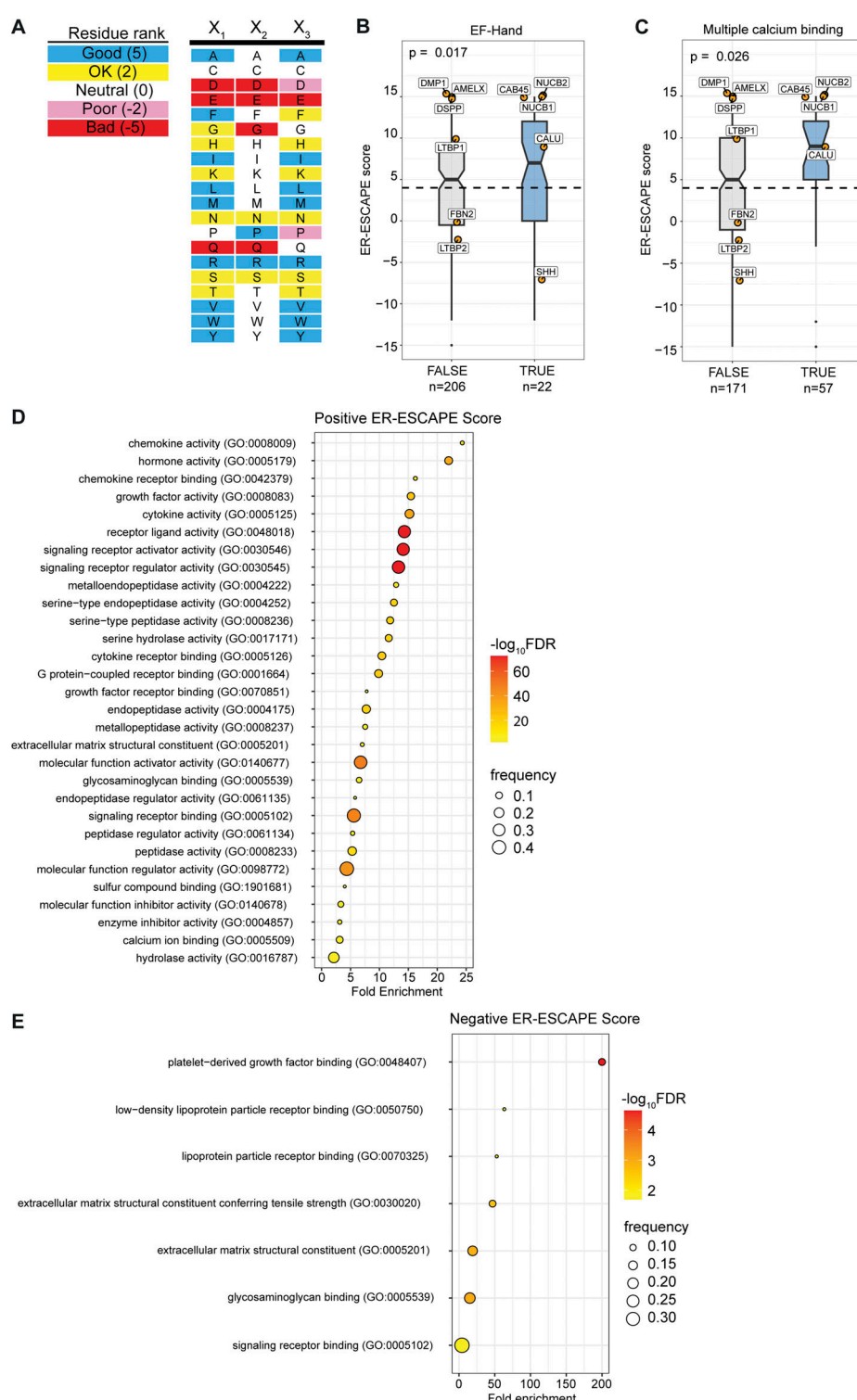

Figure S5. **ER-ESCAPE score determination and comparison of across protein properties, and Gene Ontology (GO) analysis. (A)** Scoring matrix for calculating ER-ESCAPE score. **(B)** Among annotated calcium binding proteins in the curated soluble protein secretome, proteins with annotated EF-hand domains (PROSITE) have a significantly higher ER-ESCAPE score than those without EF-Hand annotation. **(C)** Among annotated calcium binding proteins in the curated soluble protein secretome, proteins with multiple calcium binding sites (according to UniProt binding site annotation) have a significantly higher ER-ESCAPE score than those with a single calcium binding site. Curated SURF4 cargoes are highlighted as orange dots with gene names. In each plot, the dashed line represents ER-ESCAPE score median of the whole dataset ($n = 1988$), boxes represent interquartile range, whiskers represent ranges of distribution, and notches represent 95% confidence interval of the median. Kruskal–Wallis test is used in each plot to test the significance and calculate P value. Sample size is annotated in each plot below class labels. **(D and E)** GO enrichment analysis for (D) high positive ER-ESCAPE score (ER-ESCAPE score = 15 or 12), or (E) negative ER-ESCAPE score (ER-ESCAPE score = −15 or −12). In each case, GO terms are sorted by fold enrichment, colored by negative logarithm of false discovery rate (FDR); the size of each dot corresponds to the frequency of observation for each term among the high-positive or negative group.

Provided online are Table S1 and Data 1. Table S1 contains three sheets: a summary of SURF4 mutants and their phenotypes; oligonucleotide sequences used for site-directed mutagenesis and cloning; and information on commercial siRNA reagents used in knock down experiments. Data S1 contains information used in the calculation and analysis of ER-ESCAPE scores presented in Figs. 6 and S5.

