## [Peer Review File · The Journal of Cell Biology]

ER export via SURF4 uses diverse mechanisms of both client and coat engagement

Julija Maldutyte, Xiao-Han Li, Natalia Gomez-Navarro, Evan Robertson, and Elizabeth Miller

Corresponding Author(s): Elizabeth Miller, MRC Laboratory of Molecular Biology

Review Timeline:

Submission Date:	2024-06-17
Editorial Decision:	2024-07-24
Revision Received:	2024-09-17
Editorial Decision:	2024-10-01
Revision Received:	2024-10-08

Monitoring Editor: Jodi Nunnari

Scientific Editor: Andrea Marat

Transaction Report:

DOI: <https://doi.org/10.1083/jcb.202406103>

July 24, 2024

Re: JCB manuscript #202406103

Dr. Elizabeth A Miller
MRC Laboratory of Molecular Biology
Francis Crick Avenue
Cambridge CB2 0QH
United Kingdom

Dear Liz,

Thank you for submitting your manuscript entitled "ER export via SURF4 uses diverse mechanisms of both client and coat engagement". The manuscript was assessed by expert reviewers, whose comments are appended to this letter. We invite you to submit a revision if you can address the reviewers' key concerns, as outlined here.

You will see that the reviewers are overall very positive about the quality of your data and potential impact of your study. They have provided constructive feedback, most notably improve the clarity to ensure the significance of your findings are accessible to a broad readership. Most of their points can be addressed with text edits and explanations, but we encourage you to consider their minor experimental suggestions and address those with additional data at your discretion. Otherwise, please completely respond to all reviewer points in your revised study.

GENERAL GUIDELINES:

Text limits: Character count for an Article is < 40,000, not including spaces. Count includes title page, abstract, introduction, results, discussion, and acknowledgments. Count does not include materials and methods, figure legends, references, tables, or supplemental legends.

Figures: Articles may have up to 10 main text figures. Figures must be prepared according to the policies outlined in our Instructions to Authors, under Data Presentation, <https://jcb.rupress.org/site/misc/ifora.xhtml>. All figures in accepted manuscripts will be screened prior to publication.

Supplemental information: There are strict limits on the allowable amount of supplemental data. Articles may have up to 5 supplemental figures. Up to 10 supplemental videos or flash animations are allowed. A summary of all supplemental material should appear at the end of the Materials and methods section.

Please note that JCB now requires authors to submit Source Data used to generate figures containing gels and Western blots with all revised manuscripts. This Source Data consists of fully uncropped and unprocessed images for each gel/blot displayed in the main and supplemental figures. Since your paper includes cropped gel and/or blot images, please be sure to provide one Source Data file for each figure that contains gels and/or blots along with your revised manuscript files. File names for Source Data figures should be alphanumeric without any spaces or special characters (i.e., SourceDataF#, where F# refers to the associated main figure number or SourceDataFS# for those associated with Supplementary figures). The lanes of the gels/blots should be labeled as they are in the associated figure, the place where cropping was applied should be marked (with a box), and molecular weight/size standards should be labeled wherever possible. Source Data files will be made available to reviewers during evaluation of revised manuscripts and, if your paper is eventually published in JCB, the files will be directly linked to specific figures in the published article.

The typical timeframe for revisions is three to four months. While most universities and institutes have reopened labs and allowed researchers to begin working at nearly pre-pandemic levels, we at JCB realize that the lingering effects of the COVID-19 pandemic may still be impacting some aspects of your work, including the acquisition of equipment and reagents. Therefore,

if you anticipate any difficulties in meeting this aforementioned revision time limit, please contact us and we can work with you to find an appropriate time frame for resubmission. Please note that papers are generally considered through only one revision cycle, so any revised manuscript will likely be either accepted or rejected.

Thank you for this interesting contribution to Journal of Cell Biology. You can contact us at the journal office with any questions at cellbio@rockefeller.edu.

Sincerely,

Jodi Nunnari, Ph.D.
Editor-in-Chief

Andrea L. Marat, Ph.D.
Deputy Editor

Journal of Cell Biology

Reviewer #1 (Comments to the Authors (Required)):

Maldutyte et. al present strong evidence for a complex interaction of sorting co-receptors in the capture of selected secretory cargo proteins into COPII vesicles in cultured human cells, and for the decoding of these complex signals by different paralogs of the Sec24 subunit of the COPII coat. I found the experiments persuasive that the Sec24A/B pair and the Sec24C/D pair use different sorting surfaces to recognize distinct luminal loops of the SURF4 sorting receptor. I was also intrigued to learn that SURF4 employs a co-receptor, TMED10, for the recognition of the cargo molecule PCSK9 for which SURF4 has been shown to be required for ER sorting. But perhaps the most original finding presented in this work is that SURF4 may recognize some sorting cargo during translation and translocation into the lumen of the ER. Here as well, I found the data persuasive that Cab45 is recognized and co-translationally captured by SURF4, mediated by an interaction with an ER escape signal directly adjacent to the signal peptide cleavage site.

I have only two questions that the authors should be able to address in a response to the editor and a few small suggestions that can be handled by edits of the text.

1. The authors show that the GOLD but not the coiled-coil domain of TMED10 is required for sorting of PCSK9. They conclude that the GOLD domain is the binding site on TMED10 required for the capture of the immature precursor form of PCSK9 and sorting by SEC24A. In the Discussion they suggest that pro-peptide processed PCSK9 interacts with oligomeric TMED10. However, as they point out, the coiled-coil domain of TMED10, which is required for the formation of homo- or heterooligomers of the TMED paralogs, is not required for interaction of PCSK9 precursor with TMED10. Are the authors suggesting two independent and sequential interactions between PCSK9 and TMED10? One, before processing, dependent on TMED10 monomer, and the other, after processing, dependent on oligomer and SURF4?
2. The evidence that Cab45 is captured by SURF4 co-translationally is intriguing. Could it be that SURF4 is serving purely as a chaperone and not as a sorting receptor for Cab45? The authors further demonstrate that an inhibitor of signal peptide cleavage blocks that interaction. What is the rate of secretion of unprocessed Cab45 in cells treated with the signal peptidase inhibitor? What is the evidence that Cab45 folds properly in the lumen of the ER in SURF4 knock-out cells?

Minor quibbles:

1. The authors refer to the use of semi-permeabilised cells for their cell-free translation-translocation and photo-crosslinking experiments. What does it mean to be semi-permeabilised? These cells are permeable to proteins and the components for translation. Why not just say permeabilised?
2. Use sediment or centrifuge and not "pellet" or "pelleted" cells. Also, sentences should not begin with a numeral unless it is spelled out.

Reviewer #2 (Comments to the Authors (Required)):

This manuscript reports a detailed analysis of SURF4 function as an ER cargo receptor and its interaction with three specific putative cargoes, PCSK9, CAB45, and NUCB1. Evidence is presented suggesting unique interactions of these cargoes with the SEC24 paralogs, with PCSK9 more dependent on SEC24A and CAB45 and NUCB1 on SEC24C/D. In addition, a potential coreceptor for SURF4 interaction with PCSK9 is identified, TMED10. Additional complex bioinformatic analyses are presented in an effort to identify unique classes of proteins dependent on SURF4 and the specific signals used for these interactions. Overall, the data are of high quality and an impressive number of experiments and volume of data are presented. However, many of the results are highly technical and sometimes difficult to follow, and additional detail and explanation would be helpful. A major area for improvement would be to make the manuscript more readable and accessible to the typical JCB reader, with its current highly technical form likely only fully intelligible to a very limited number of experts in this specific subfield.

Specific comments:

1. The authors report dependence of CAB45 secretion on SEC24C/D. Data in Figure 1 (Panel D) demonstrate that, while knockout of SEC24D alone has no effect on CAB45 secretion, simultaneous knockout of SEC24D, together with knockdown of SEC24C, does significantly impede CAB45 secretion. However, SEC24C alone was not studied. Could SEC24C be the key SEC24 paralog by itself, with no contribution from SEC24D? Also, how effective was the SEC24C si knockdown?
2. The specific SEC24A mutants at the B, IxM and D site are hard to tease out. Are these based on previous studies? What are the "Phe loop" and "DE" mutants? Can indirect effects on other parts of the SEC24A structure be excluded? Similarly, multiple SURF4 mutants are studied, and it's difficult to follow why these specific substitutions were chosen, and again, can indirect conformational effects on other parts of the molecule be excluded? Also, what is the "PRKACA" negative control in Fig 1-3?
3. The authors don't clearly explain why the DepMap and co-essentiality analysis should be relevant for identifying co-receptors for SURF4, and the analysis shown in Figure 3A is not clear to someone unfamiliar with these methods. What are the X and Y axes and significance of adjacent points?
4. Several deletion mutants for TMED10 are studied for their effect on interaction with PCSK9, with only deletion of the GOLD domain interfering with PCSK9 binding. Can indirect effects of this deletion on the conformation of other segments of TMED10 be excluded? Does a recombinant isolated GOLD domain retain PCSK9 binding activity?

Minor points:

- 1) The term "paralogous" is loosely applied. Technically, the yeast and human SEC24s should probably be referred to as "orthologs" and SEC24A, B, C, and D are all paralogs, not just C with D and A with B.
- 2) Page 15, Cloning subsection of Methods: "closed" should be "cloned".
- 3) The SURF4 AlphaFold2 structure in Figure 2A shows 8 transmembrane domains, whereas the cartoons in Figs 1E and 2D show 4.

Reviewer #3 (Comments to the Authors (Required)):

Maldutyte and colleagues address the important cell biological question how the interplay of sorting signals, cargo receptors and vesicle coat proteins at the ER enables cells to (1) sort newly synthesized secretory proteins from ER-resident proteins and (2) coordinate the maturation of secretory proteins with their ER export. Building on extensive previous work, they dissect the recognition of secretory cargos by the ER cargo receptor, SURF4, and the SEC24 cargo adaptor of the COPII coat in mammalian cells. They show that different soluble cargo proteins, which all engage SURF4, use different SEC24 isoforms. The authors explain this unexpected selectivity by showing that certain cargo proteins additionally use the TMED10 co-receptor and thereby trigger a different mode of interaction between cargo receptors and SEC24. The authors carefully define binding determinants in SURF4, TMED10 and SEC24 to provide strong support for this model. In the second part of the paper, the authors show - by technically demanding photo-crosslinking experiments - that cargo proteins can bind to SURF4 co-translationally, before they have fully folded and matured. This is again an unexpected result that opens up many new and interesting questions. The authors complement these findings with bioinformatics analyses and suggest that the ER transit time of secretory proteins is determined, at least in part, by their affinity for SURF4, resulting in a "fast track" for certain proteins.

This is a very clear, technically impressive and well-crafted study that adds fresh and surprising information to our understanding of a fundamental biological process. The manuscript is a perfect fit for JCB and will certainly be appreciated by its readers. The only additional experiment I would suggest is to ask directly if TMED10 indeed binds to the B site of SEC24A. Other than that, only minimal revision is needed (see minor comments below).

Minor comments

1. P3, last sentence of introduction: It is not immediately clear what is meant with "Ca²⁺ flux". Could 'homeostasis' perhaps be an alternative?
2. Figure 2C: Can you please explain the blue/purple color code for the mutations on the x-axis? It would be helpful if it were immediately obvious which mutations are aimed at disrupting the B and D sites, respectively.
3. P5: Figure S4A is not called out in the text. Furthermore, Figures S4C, D come up before Figure S4B, which is odd.

4. Figure S4A: the label is shifted relative to the lanes in the panel on the right.
5. P6: "In contrast, no interaction was detected when the GOLD domain was deleted (Fig. 3E)". IP of TMED10 lacking the GOLD domain still brings down PCSK9, this time predominantly the mature form. Can the author please offer an explanation?
6. The authors imply but do not show that TMED10 binds to the B site of SEC24A. For completeness' sake, it would be nice to test this directly. This could be done by looking at the interaction of TMED10 with wild-type SEC24A and a B-site mutant or by asking whether loss of TMED10 still matters for the interaction of SURF4 and a SEC24A B-site mutant (similar to the experiment shown in Figure 2C).
7. Figure 3: The cartoon in Figure 2D is useful to help the reader keep track of who interacts with whom how. Inserting a similar (and refined) cartoon at the end of Figure 3 would be very helpful, too.
8. Figure 4B: Three forms of Cab45 are indicated on the right but it is difficult to see which bands they correspond to.
9. P7, last sentence: "In contrast, cross-links ... unaffected." This sentence sounds a bit odd. Perhaps it would be clearer to say something like: In contrast, cross-links were unaffected in a more distant luminal mutant, ... CW binding site mutant, EDD.
10. Figure 5: Please indicate in the figure legend for panel A what PK stands for. Also indicate that panel A and B deal with Cab45.
11. P9: "... it's previously been proposed ...". Please change it's to it has.
12. P10: "... might predict fast transit times". Shouldn't this be 'short' transit times?
13. P11, top: Again, it is not obvious (at least to me) what is meant by calcium "flux".
14. P12: "... identified 747 proteins with at least one CW motif (Supplemental Data)". Which Supplemental data are the authors referring to? Please give an exact reference.

Reviewer 1

Major points:

1. The authors show that the GOLD but not the coiled-coil domain of TMED10 is required for sorting of PCSK9. They conclude that the GOLD domain is the binding site on TMED10 required for the capture of the immature precursor form of PCSK9 and sorting by SEC24A. In the Discussion they suggest that pro-peptide processed PCSK9 interacts with oligomeric TMED10. However, as they point out, the coiled-coil domain of TMED10, which is required for the formation of homo- or heterooligomers of the TMED paralogs, is not required for interaction of PCSK9 precursor with TMED10. Are the authors suggesting two independent and sequential interactions between PCSK9 and TMED10? One, before processing, dependent on TMED10 monomer, and the other, after processing, dependent on oligomer and SURF4?

We apologize for the lack of clarity. The co-IP experiment using TMED10 deletion mutants demonstrates that PCSK9 interacts with TMED10 via GOLD domain, which is known to interact with other cargoes. In the Discussion, we simply aimed to make the point that p24 proteins are almost always oligomeric, so we expect TMED10 to form a homo- or hetero-tetramer. The coiled-coil deletion experiment emphasizes that PCSK9 interaction is only via TMED10, regardless of its partner p24 proteins. We do not yet know whether CC-deleted TMED10, which still interacts with PCSK9, is competent for PCSK9 export. This is an interesting experiment that will be explored in the future, along with whether other p24 partner proteins also contribute to export independent of direct binding to PCSK9. We have tightened up the text surrounding this model in the relevant Results section (lines 191-200).

2. The evidence that Cab45 is captured by SURF4 co-translationally is intriguing. Could it be that SURF4 is serving purely as a chaperone and not as a sorting receptor for Cab45?

This is an interesting suggestion, and one we have considered philosophically: what defines a chaperone, a “holdase”, an “escort”, a receptor? SURF4 itself leaves the ER so doesn't behave like a canonical chaperone to bind and release its clients for multiple rounds of folding. We consider it more of a holdase AND escort: it binds (perhaps preventing inappropriate interactions but unlikely to directly stimulate folding) and contributes to ER export (via its own ER export signals). For other clients, like PCSK9, a more direct escort/receptor function is likely. Understanding exactly how client binding influences Sec24 binding likely holds the key to some of these functional definitions, and awaits more precise biochemical and structural studies.

The authors further demonstrate that an inhibitor of signal peptide cleavage blocks that interaction. What is the rate of secretion of unprocessed Cab45 in cells treated with the signal peptidase inhibitor?

We have now conducted a pulse-chase experiment where we compare Cab45-HA secretion for WT and SP cleavage mutants (new Figure S5F). Indeed, SP mutant secretion was minimal, consistent with an inability to bind SURF4 for export.

What is the evidence that Cab45 folds properly in the lumen of the ER in SURF4 knock-out cells?

Firstly, the ER-retained protein is stable and not degraded, which would be expected of a misfolded form. Secondly, we tested for Cab45 aggregation in SURF4 KO cells using native gels and did not observe any higher order species. These lines of evidence, combined with the lack of an activated UPR in the SURF4 KO cells lead us to conclude that Cab45 is unlikely to be dramatically misfolded.

Minor quibbles:

1. The authors refer to the use of semi-permeabilised cells for their cell-free translation-translocation and photo-crosslinking experiments. What does it mean to be semi-permeabilised? These cells are permeable to proteins and the components for translation. Why not just say permeabilised?

This is a good question. The cells are treated with digitonin, which presumably permeabilizes the plasma membrane but leaves other organelles intact. For example, the nucleus and ER membranes remain impermeable to proteins. Since this seems to be the term that the field uses for *in vitro* translation experiments, we have retained this term.

2. Use sediment or centrifuge and not "pellet" or "pelleted" cells. Also, sentences should not begin with a numeral unless it is spelled out.

Fixed.

Reviewer 2

A major area for improvement would be to make the manuscript more readable and accessible to the typical JCB reader, with its current highly technical form likely only fully intelligible to a very limited number of experts in this specific subfield.

We have endeavoured to simplify the language and avoid technical terms where possible, while still providing necessary experimental detail.

Specific comments:

1. The authors report dependence of CAB45 secretion on SEC24C/D. Data in Figure 1 (Panel D) demonstrate that, while knockout of SEC24D alone has no effect on CAB45 secretion, simultaneous knockout of SEC24D, together with knockdown of SEC24C, does significantly impede CAB45 secretion. However, SEC24C alone was not studied. Could SEC24C be the key SEC24 paralog by itself, with no contribution from SEC24D? Also, how effective was the SEC24C si knockdown?

We have indeed tested the SEC24C single mutant but in the interest of space didn't include it in the original submission. We now show Cab45-HA and NUCB1-HA secretion in WT and SEC24C KO cells, where there is no difference between the two cell lines. The single KO cells (SEC24B, C and D) have all been moved to the supplement (Fig S1A), allowing to focus on the A/B and C/D double mutants in the main figure.

The SEC24C si KD is very efficient (see Reviewer Figure below). We didn't include this analysis since the effect on secretion is very clear: even if KD is not fully penetrant, we still see a clear secretion phenotype, which is not the case with SED24D KO alone.

2. The specific SEC24A mutants at the B, IxM and D site are hard to tease out. Are these based on previous studies? What are the "Phe loop" and "DE" mutants? Can indirect effects on other parts of the SEC24A structure be excluded? Similarly, multiple SURF4 mutants are studied, and it's difficult to follow why these specific substitutions were chosen, and again, can indirect conformational effects on other parts of the molecule be excluded? Also, what is the "PRKACA" negative control in Fig 1-3?

Apologies for not making this clearer. Yes, most of the SEC24 mutants have been previously characterized, largely by the Goldberg lab using precise biochemical and structural studies. The D-site is the only one that hasn't been characterized for human SEC24, but we designed mutations based on our well-characterized yeast mutant and structural/sequence homology. In general, indirect effects on SEC24 are unlikely: previous characterization of yeast mutants shows that mutation of one cargo binding site does not impact capture of cargo that use different sites. Moreover, structural analysis of SEC24 suggests that the structure itself is very robust, well-folded and unlikely to undergo allosteric change. We've now included structures that show exactly where these mutations lie and have cited the relevant papers. We have also amended the NanoBiT graphs - we have converted these to bar graphs with the bars colour coded to match mutation sites indicated in the SEC24A and SEC24C structures (Figures 1 and 2, respectively).

The SURF4 mutants (including the Phe-loop and DE mutants) arose from a systematic mutational screen of conserved residues. The full panel of mutants, along with their stability and secretion phenotypes are presented in Figure S2. For further analysis, we chose mutants that were stably expressed (and thus unlikely to be misfolded) but had clear secretion defects for Cab45. We have tried to explain this in more detail so the reader can follow our rationale (lines 119-132).

Finally, the PRKACA negative control is a luciferase fusion to an irrelevant protein that serves to control for luciferase-driven affinity. We've re-labelled these and included the explanation in the figure legend.

3. The authors don't clearly explain why the DepMap and co-essentiality analysis should be relevant for identifying co-receptors for SURF4, and the analysis shown in Figure 3A is not clear to someone unfamiliar with these methods. What are the X and Y axes and significance of adjacent points?

We now explain the basis for DepMap and co-essentiality analyses and have moved the figure to the supplement. It is indeed confusing since this is not a graph with axes per se but a representation of nearest-neighbours in a large protein-protein interaction network. More detail is included in the figure legend to explain this.

4. Several deletion mutants for TMED10 are studied for their effect on interaction with PCSK9, with only deletion of the GOLD domain interfering with PCSK9 binding. Can indirect effects of this deletion on the conformation of other segments of TMED10 be excluded? Does a recombinant isolated GOLD domain retain PCSK9 binding activity?

Although we can't definitively rule out indirect effects of the mutations we made, previous structural, biochemical and mutational analysis is all consistent with our interpretations. The GOLD domain deletion should not affect the conformation of the coiled-coiled domain, since the coiled-coil is known to be a rigid structure. It's been well-established that TMED proteins homo- and hetero-oligomerise via their coiled-coil domains, so the CC mutant is almost certainly not in complex with other family members. The fact that we still see interaction with PCSK9 suggests that other TMED proteins do not contribute to the interaction. We have not tested whether recombinant TMED10 GOLD domain binds PCSK9 in vitro. This would be an excellent tool to study conditional binding in the context of PCSK9 cleavage and SURF4 engagement, and will certainly be an avenue for future studies.

Minor points:

1) The term "paralogous" is loosely applied. Technically, the yeast and human SEC24s should probably be referred to as "orthologs" and SEC24A, B, C, and D are all paralogs, not just C with D and A with B.

We have tried to be more precise in our terminology throughout.

2) Page 15, Cloning subsection of Methods: "closed" should be "cloned".

Fixed.

3) The SURF4 AlphaFold2 structure in Figure 2A shows 8 transmembrane domains, whereas the cartoons in Figs 1E and 2D show 4.

Cartoons have been fixed to show 8 transmembrane domains.

Reviewer 3

The only additional experiment I would suggest is to ask directly if TMED10 indeed binds to the B site of SEC24A. Other than that, only minimal revision is needed (see minor comments below).

We have attempted to address this by two methods: we tested a TMED10 NanoBiT fusion for interaction with SEC24A, and we tested cross-linking and co-IP to detect interaction similar to that used for the PCSK9 interaction. Neither experiment was successful in capturing an interaction between TMED10 and SEC24A, making the B-

site analysis impossible. We have added a description of this in the Results (lines 198-204):

We were unable to directly test TMED10 engagement at the B-site of SEC24A; LgBiT-TMED10 did not function in the NanoBiT assay, presumably because the luciferase construct sterically occludes its ER export signal. Moreover, we could not detect an interaction by crosslinking co-IP, likely due to the transient nature of TMED10-SEC24A interaction. Based on previous detailed biochemical and structural studies (Ma et al., 2017; Nie et al., 2018; Nufer et al., 2002; Strating and Martens, 2009) we propose that TMED10 is likely to engage the SEC24A B-site via either its FF or C-terminal IE motif.

Minor comments

1. P3, last sentence of introduction: It is not immediately clear what is meant with "Ca²⁺ flux". Could 'homeostasis' perhaps be an alternative?

Fixed.

2. Figure 2C: Can you please explain the blue/purple color code for the mutations on the x-axis? It would be helpful if it were immediately obvious which mutations are aimed at disrupting the B and D sites, respectively.

Fixed. Graphs were changed to bar graphs and colour-coded to match mutation sites indicated in SEC24A crystal structure that was moved to the main figure from the supplement.

3. P5: Figure S4A is not called out in the text. Furthermore, Figures S4C, D come up before Figure S4B, which is odd.

Fixed.

4. Figure S4A: the label is shifted relative to the lanes in the panel on the right.

Fixed.

5. P6: "In contrast, no interaction was detected when the GOLD domain was deleted (Fig. 3E)". IP of TMED10 lacking the GOLD domain still brings down PCSK9, this time predominantly the mature form. Can the author please offer an explanation?

We have added a section in the Results that explains our thinking (lines 191-196):

In contrast, upon GOLD domain deletion, TMED10 recovery with V5-PCSK9 was dramatically reduced. In the reciprocal IP, the precursor form of PCSK9 was poorly recovered, and instead the mature form was more abundant (Fig. 3D). One possibility is that the mature PCSK9 recovered is bound to SURF4, which is presumably within the holo-complex that drives ER export and therefore also recovered in the TMED10 IP.

6. The authors imply but do not show that TMED10 binds to the B site of SEC24A. For completeness' sake, it would be nice to test this directly. This could be done by

looking at the interaction of TMED10 with wild-type SEC24A and a B-site mutant or by asking whether loss of TMED10 still matters for the interaction of SURF4 and a SEC24A B-site mutant (similar to the experiment shown in Figure 2C).

As briefly described above, we attempted to show an interaction between TMED10 and SEC24A that would permit dissection of the binding interaction. Our attempts to use TMED10 in the NanoBIT assay fused LgBiT to the short cytosolic tail of TMED10, which unfortunately yielded no signal. This is likely due to the steric effects caused by the size of the LgBiT domain (18 kDa) that would preclude interaction with SEC24, especially if the interaction is via the well-characterized B-site mode of binding where a small helical or C-terminal peptide inserts into the binding pocket.

The suggestion to test the effect of TMED10 KD on the interaction between SURF4 and the SEC24A B-site mutant is unlikely to yield a robust result since B-site mutations on their own cause a large reduction in signal (Fig. 2D).

To further explore the interaction between TMED10 and SEC24A, we tried a chemical crosslinking (with DSP and SMPT) and co-IP approach, similar to the PCSK9 and TMED10 interaction experiments. Unfortunately, an interaction between WT TMED10 and WT SEC24A could not be detected since it is likely too transient, i.e. a receptor interaction with the inner COPII coat is even shorter lived than that between a receptor and cargo inside the ER and vesicle. Nevertheless, the amino acid sequence of the TMED10 cytosolic tail and previous structural and biochemical studies strongly suggest TMED10 interaction at SEC24A B-site: TMED10 contains both an FF and an acidic IE C-terminal motifs, both of which have been shown to engage the B-site (Ma et al., 2017; Nie et al., 2018; Nufer et al., 2002; Strating and Martens, 2009). This explanation is now included in the text (p6-7).

7. Figure 3: The cartoon in Figure 2D is useful to help the reader keep track of who interacts with whom how. Inserting a similar (and refined) cartoon at the end of Figure 3 would be very helpful, too.

Cartoon added as suggested.

8. Figure 4B: Three forms of Cab45 are indicated on the right but it is difficult to see which bands they correspond to.

Apologies – we agree that these bands are difficult to parse. We've simplified this now to just indicate the glycosylated and unglycosylated species, which are the relevant features. More detailed (and clearer) bands are shown in the supplemental figure that demonstrates proteinase K protection.

9. P7, last sentence: "In contrast, cross-links ... unaffected." This sentence sounds a bit odd. Perhaps it would be clearer to say something like: In contrast, cross-links were unaffected in a more distant luminal mutant, ... CW binding site mutant, EDD.

Fixed.

10. Figure 5: Please indicate in the figure legend for panel A what PK stands for. Also indicate that panel A and B deal with Cab45.

Fixed.

11. P9: "... it's previously been proposed ...". Please change it's to it has.

Fixed.

12. P10: "... might predict fast transit times". Shouldn't this be 'short' transit times?

Fixed.

13. P11, top: Again, it is not obvious (at least to me) what is meant by calcium "flux".

Fixed.

14. P12: "... identified 747 proteins with at least one CW motif (Supplemental Data)". Which Supplemental data are the authors referring to? Please give an exact reference.

Fixed.

October 1, 2024

RE: JCB Manuscript #202406103R

Dr. Elizabeth A Miller
MRC Laboratory of Molecular Biology
Francis Crick Avenue
Cambridge CB2 0QH
United Kingdom

Dear Liz,

Thank you for submitting your revised manuscript entitled "ER export via SURF4 uses diverse mechanisms of both client and coat engagement". We would be happy to publish your paper in JCB pending final revisions necessary to meet our formatting guidelines (see details below).

A. MANUSCRIPT ORGANIZATION AND FORMATTING:

- 1) Text limits: Character count for Articles is < 40,000, not including spaces. Count includes abstract, introduction, results, discussion, and acknowledgments. Count does not include title page, figure legends, materials and methods, references, tables, or supplemental legends.
- 2) Figures limits: Articles may have up to 10 main text figures.
- 3) Figure formatting: Scale bars must be present on all microscopy images, including inset magnifications (you may alternatively indicate the diameter of the inset). Molecular weight or nucleic acid size markers must be included on all gel electrophoresis. Aspect ratios of images may not be altered.
- 4) Statistical analysis: Error bars on graphic representations of numerical data must be clearly described in the figure legend. The number of independent data points (n) represented in a graph must be indicated in the legend. Statistical methods should be explained in full in the materials and methods. For figures presenting pooled data the statistical measure should be defined in the figure legends. Please also be sure to indicate the statistical tests used in each of your experiments (either in the figure legend itself or in a separate methods section) as well as the parameters of the test (for example, if you ran a t-test, please indicate if it was one- or two-sided, etc.). Also, if you used parametric tests, please indicate if the data distribution was tested for normality (and if so, how). If not, you must state something to the effect that "Data distribution was assumed to be normal but this was not formally tested."
- 5) Abstract and title: The abstract should be no longer than 160 words and should communicate the significance of the paper for a general audience. The title should be less than 100 characters including spaces. Make the title concise but accessible to a general readership.
- 6) Materials and methods: Should be comprehensive and not simply reference a previous publication for details on how an experiment was performed. Please provide full descriptions in the text for readers who may not have access to referenced manuscripts.
- 7) All antibodies, cell lines, animals, and tools used in the manuscript should be described in full, including accession numbers for materials available in a public repository such as the Resource Identification Portal. Please be sure to provide the sequences for all of your primers/oligos and RNAi constructs in the materials and methods. You must also indicate in the methods the source, species, and catalog numbers (where appropriate) for all of your antibodies. Please also indicate the acquisition and quantification methods for immunoblotting/western blots.
- 8) Microscope image acquisition: The following information must be provided about the acquisition and processing of images:
 - a. Make and model of microscope
 - b. Type, magnification, and numerical aperture of the objective lenses
 - c. Temperature
 - d. Imaging medium

- e. Fluorochromes
- f. Camera make and model
- g. Acquisition software
- h. Any software used for image processing subsequent to data acquisition. Please include details and types of operations involved (e.g., type of deconvolution, 3D reconstitutions, surface or volume rendering, gamma adjustments, etc.).

10) Supplemental materials: There are strict limits on the allowable amount of supplemental data. Articles may have up to 5 supplemental figures. Please reduce your SI count and correct the callouts in the text to reflect any changes. Please also note that tables, like figures, should be provided as individual, editable files. A summary of all supplemental material should appear at the end of the Materials and methods section.

13) ORCID IDs: ORCID IDs are unique identifiers allowing researchers to create a record of their various scholarly contributions in a single place. Please note that ORCID IDs are now *required* for all authors. At resubmission of your final files, please be sure to provide your ORCID ID and those of all co-authors.

Please note that JCB now requires authors to submit Source Data used to generate figures containing gels and Western blots with all revised manuscripts. This Source Data consists of fully uncropped and unprocessed images for each gel/blot displayed in the main and supplemental figures. Since your paper includes cropped gel and/or blot images, please be sure to provide one Source Data file for each figure that contains gels and/or blots along with your revised manuscript files. File names for Source Data figures should be alphanumeric without any spaces or special characters (i.e., SourceDataF#, where F# refers to the associated main figure number or SourceDataFS# for those associated with Supplementary figures). The lanes of the gels/blots should be labeled as they are in the associated figure, the place where cropping was applied should be marked (with a box), and molecular weight/size standards should be labeled wherever possible. Source Data files will be made available to reviewers during evaluation of revised manuscripts and, if your paper is eventually published in JCB, the files will be directly linked to specific figures in the published article.

Journal of Cell Biology now requires a data availability statement for all research article submissions. These statements will be published in the article directly above the Acknowledgments. The statement should address all data underlying the research presented in the manuscript. Please visit the JCB instructions for authors for guidelines and examples of statements at (<https://rupress.org/jcb/pages/editorial-policies#data-availability-statement>).

B. FINAL FILES:

****It is JCB policy that if requested, original data images must be made available to the editors. Failure to provide original images upon request will result in unavoidable delays in publication. Please ensure that you have access to all original data images prior to final submission.****

****The license to publish form must be signed before your manuscript can be sent to production. A link to the electronic license to publish form will be sent to the corresponding author only. Please take a moment to check your funder requirements before choosing the appropriate license.****

Thank you for your attention to these final processing requirements. Please revise and format the manuscript and upload materials within 7 days. If you need an extension for whatever reason, please let us know and we can work with you to determine a suitable revision period.

Thank you for this interesting contribution, we look forward to publishing your paper in Journal of Cell Biology.

Sincerely,

Jodi Nunnari, Ph.D.
Editor-in-Chief

Andrea L. Marat, Ph.D.
Deputy Editor

Journal of Cell Biology